# Sirtuin 2 promotes human cytomegalovirus replication by regulating cell cycle progression

Cora N. Betsinger,[1] Joshua L. Justice,[1] Matthew D. Tyl,[1] Julia E. Edgar,[1] Hanna G. Budayeva,[1] Yaa F. Abu,[1] Ileana M. Cristea[1]

**ABSTRACT**  The ability to modulate host cell cycle progression is a requirement of many human viruses in order to facilitate their replication and propagation. Nuclear-replicating DNA viruses frequently stall the host cell cycle in G1 to avoid competition with host DNA replication. Among these viruses is human cytomegalovirus (HCMV), a prevalent beta-herpesvirus. Here, we discover a pro-viral mechanism that employs the deacetylase activity of the human enzyme sirtuin 2 (SIRT2) for HCMV-mediated cell cycle dysregulation. First, we show that the SIRT2 deacetylase activity supports an early stage of HCMV replication. Focusing on these early infection time points, we next define temporal SIRT2 protein interactions and deacetylation substrates by using mass spectrometry-based interactome and acetylome analyses. We find that SIRT2 interacts with and modulates the acetylation level of cell cycle proteins during infection, including the cyclin-dependent kinase 2 (CDK2). Using flow cytometry, cell sorting, and functional assays, we demonstrate that SIRT2 regulates CDK2 K6 acetylation and the G1- to S-phase transition in a manner that supports HCMV replication. Altogether, our findings expand the understanding of mechanisms underlying HCMV-induced cell cycle dysregulation and point toward regulatory feedback between SIRT2 and CDK2 that can have implications in other viral infections and human diseases.

**IMPORTANCE**  This study expands the growing understanding that protein acetylation is a highly regulated molecular toggle of protein function in both host anti-viral defense and viral replication. We describe a pro-viral role for the human enzyme SIRT2, showing that its deacetylase activity supports HCMV replication. By integrating quantitative proteomics, flow cytometry cell cycle assays, microscopy, and functional virology assays, we investigate the temporality of SIRT2 functions and substrates. We identify a pro-viral role for the SIRT2 deacetylase activity via regulation of CDK2 K6 acetylation and the G1-S cell cycle transition. These findings highlight a link between viral infection, protein acetylation, and cell cycle progression.

**KEYWORDS**  proteomics, acetylome, acetylation, protein-protein interactions, mass spectrometry, sirtuin 2

I n response to virus infections, intertwined host defense responses and virus replication processes drive numerous alterations to host cells. These alterations are reflected in dynamic changes to the cellular transcriptome, proteome, and metabolome. Such changes are frequently linked to protein post-translational modifications (PTMs) that activate cellular networks and signaling cascades to drive rapid intra- and inter-cellular communication. Among these, protein acetylation has emerged as a versatile component of cellular regulation (1–5). Dynamic protein acetylation events underlie transcriptional responses and regulation of the cell cycle and can act as signatures of the metabolic status of a cell (1, 3, 6–9). Viral infections have been shown to induce temporal

Address correspondence to Ileana M. Cristea, icristea@princeton.edu.

I.M.C. is a shareholder of Evrys Bio (previously Forge Life Science), which has licensed sirtuin-related technology from Princeton University.

See the funding table on p. 25.

changes to protein acetylation, with finely tuned modifications reported on either host or virus proteins at different stages of a virus replication cycle (10–14).

Global changes to the cellular acetylome have been observed upon infection with human cytomegalovirus (HCMV) (11), a prevalent pathogen estimated to infect over half of the world's population (15). HCMV is the leading infectious cause of congenital birth defects, a major concern for immunocompromised individuals, and is associated with a range of chronic human diseases, including cardiovascular disease and cancer (16–21). Protein acetylation has been shown to contribute to both HCMV replication and host defense responses. For example, lamin acetylation affects virus capsid nuclear egress (11); acetylation of microtubules modulates nuclear rotation and the formation of the HCMV assembly compartment (22, 23); OPA1 acetylation regulates the mitochondria fusion-fission balance tuned during infection (24); histone acetylation at virus promoters provides control over virus gene expression (25); and the acetylation of viral transcriptional activator pUL26 inhibits virus production (11). Although HCMV-induced alterations to protein acetylation were observed in almost every subcellular compartment (11), the regulation and function of these PTMs and their pathological significance remain largely unknown.

A class of cellular proteins known to regulate protein acetylation is formed by sirtuins (SIRTs), which are evolutionary-conserved and ubiquitously expressed $NAD^+$-dependent enzymes (26–32). SIRTs function broadly at the interface between metabolism and cellular homeostasis by deacetylating their substrates, as well as by regulating a range of acylation modifications and other PTMs (33–38). SIRTs are also known to act as sensors of the environment, responding to external stimuli and pathogens and facilitating rapid cellular communication processes. Given their numerous substrates and activities, evidence is beginning to accumulate for SIRTs in both pro- and anti-viral capacities. The cytoplasmic SIRT isoform sirtuin 2 (SIRT2), for instance, was shown to promote the replication of hepatitis B virus and human immunodeficiency virus but inhibit influenza A virus replication (39–43). In the context of HCMV infection, knockdown of SIRT2 resulted in elevated virus titers (44). However, which of the SIRT2 enzymatic activities or the mechanisms responsible for this effect have not been characterized.

SIRT2 possesses the ability to remove diverse modifications from substrates, including acetylation, long-chain fatty-acyl, lipoyl, benzoyl, 4-oxononanoyl, and methacrylate groups (33, 45–53). The initial substrate identified for SIRT2 deacetylase activity was α-tubulin (54). SIRT2-catalyzed deacetylation of α-tubulin alters microtubule stability by allowing for greater conformational sampling (55). Through this activity, SIRT2 has been implicated in the regulation of cytoskeletal organization in diverse cellular contexts, including cell cycle progression and cell migration (27, 56, 57). Since this initial discovery, SIRT2 has been shown to regulate gene expression through the deacetylation of histone proteins and transcription factors such as FOXO1, FOXO3a, HIF1A, and p53 (58–61). SIRT2 also deacetylates key glycolytic and pentose phosphate pathway proteins, thereby modulating cellular metabolism (62, 63). Although SIRT2 predominantly localizes to the cytoplasm, localizations to the mitochondria and nucleus have been observed under different cellular stimuli, expanding its catalog of potential substrates (52, 64). Indeed, a recent acetylome analysis in colorectal cancer cells following SIRT2 overexpression and knockdown revealed close to 200 SIRT2 substrates broadly functioning in cytoskeletal organization, metabolic processes, cell cycle regulation, and gene expression (65).

Given its multi-functionality and localization to compartments where we observe global alterations in acetylation during HCMV infection, SIRT2 deacetylase activity may be poised for both virus manipulation and host defense processes. Here, we characterized the contribution of SIRT2 deacetylase activity to HCMV replication. We discovered that inhibition of SIRT2 deacetylase activity hinders an early stage of the HCMV replication cycle, indicating a pro-viral function for SIRT2. To understand what drives this function, we characterized SIRT2 substrates and temporal changes in its protein interactions by performing acetylome and interactome analyses during HCMV infection. Integration of these temporal data sets pointed to the association between SIRT2 and

cell cycle proteins, with both their interaction and acetylation status being modulated by the SIRT2 enzymatic activity. Indeed, we confirm that SIRT2 regulates cell cycle progression using flow cytometry-based cell cycle profiling. We uncovered a role for SIRT2 in the regulation of cyclin-dependent kinase 2 (CDK2) K6 acetylation status and the G1- to S-phase transition, where SIRT2 inhibition or constitutive expression of a CDK2 K6 acetylation mimic drives entry into S phase, a non-productive cell cycle state for HCMV replication (66, 67). Altogether, our findings add to the accumulating evidence of cell cycle regulation via acetylation, identify a previously unrecognized function for SIRT2 in regulating CDK2 acetylation and the G1-S transition, and mechanistically define a role for SIRT2 in promoting HCMV replication. As cell cycle manipulation is characteristic of nearly all viral infections and many human pathologies, these findings may have broad implications outside the context of HCMV (68–71).

## RESULTS

### Inhibition of SIRT2 deacetylase activity restricts an early stage of HCMV replication

Investigating the function of SIRT2 deacetylase activity is complicated by the numerous enzymatic activities of SIRT2. In addition to its deacetylase activity, SIRT2 possesses the ability to remove diverse modifications from substrates, including long-chain fatty-acyl, lipoyl, benzoyl, 4-oxononanoyl, and methacrylate groups (33, 45–53). Therefore, to specifically investigate the contribution of SIRT2 deacetylase activity to HCMV replication, we utilized the SIRT2 inhibitor AGK2 (Fig. 1A) (72, 73). AGK2 is a selective SIRT2 inhibitor, with slight inhibition of other SIRT proteins observed only at concentrations greater than 40 µM (72). Additionally, AGK2 has recently been demonstrated as specific for inhibiting SIRT2 deacetylase activity, showing no impact on SIRT2 defatty-acylase functions (73).

We first validated the use of AGK2 in our fibroblast cell culture model during HCMV infection. Unless otherwise specified, MRC5 fibroblast cells were pre-treated with AGK2 (or DMSO, vehicle control) for 12 hours (Fig. 1A). The cells were then either cultured in Dulbecco's modified Eagle medium (DMEM) alone to obtain uninfected (mock) controls or, for infection, in DMEM containing HCMV inoculum. After 1 hour, which allowed for virus absorption and entry, the media were replaced with AGK2- or DMSO-containing media, and the time points of infection were initiated with 0 hour post infection (HPI) (Fig. 1A), as previously described (74–77). We confirmed that AGK2 treatment resulted in limited cytotoxicity during HCMV infection (Fig. S1A). Additionally, using Western blotting and confocal microscopy, we observed that the acetylation status of tubulin K40 and histone H3 K9, known SIRT2 deacetylase substrates (54, 78), increased following AGK2 treatment (Fig. S1B through E). Using a combination of Western blotting and targeted mass spectrometry, we analyzed the impact of AGK2 treatment on SIRT2 protein abundance throughout HCMV infection (Fig. S1F and G). We found SIRT2 protein levels to increase slightly upon AGK2 treatment in uninfected cells and at late infection time points (i.e., 96 HPI), with no significant change in SIRT2 protein levels at 12, 24, 48, or 72 HPI for AGK2-treated samples relative to DMSO.

Having validated AGK2 efficacy in our cell culture model, we next investigated the impact of SIRT2 inhibition on HCMV replication by quantifying infectious virus particles produced from AGK2-treated cells. AGK2 treatment resulted in a 10- to 14-fold decrease in virus production during infection with either a laboratory-adapted HCMV strain (AD169) or a lower-passage and more clinically relevant strain (TB40E) (Fig. 1B). These results suggest that SIRT2 deacetylase activity supports HCMV replication.

Confocal microscopy analysis showed that the endogenous SIRT2 localizes to both the cytoplasm and the nucleus in uninfected cells and throughout infection (Fig. 1C). The HCMV replication cycle proceeds over 5 days and includes temporally orchestrated events initiating with (i) viral entry into host cells and proceeding with (ii) replication of the viral genome, (iii) assembly of new viral particles, and (iv) release of infectious progeny virions from the cell (Fig. 1C). The stages of this replication cycle take place

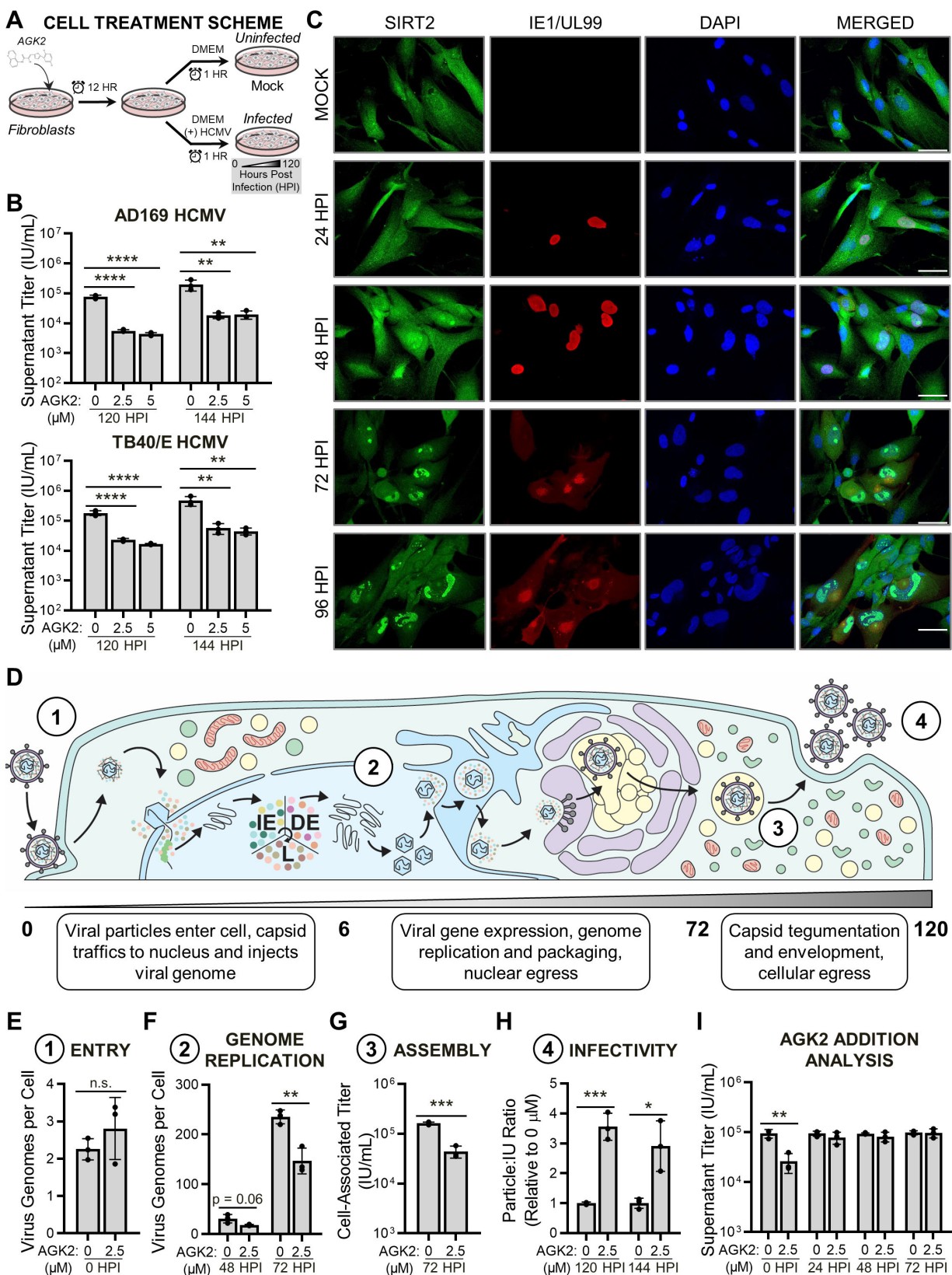

**FIG 1** SIRT2 deacetylase activity supports an early stage of the HCMV replication cycle. (A) Diagram of the AGK2 treatment and HCMV infection procedure used throughout this study. Unless otherwise indicated, fibroblasts were treated with AGK2 (or DMSO, vehicle control) 12 hours prior to infection with HCMV. Cells were then incubated in either complete growth media (for uninfected mock samples) or in complete growth media containing HCMV inoculum (for

**FIG 1** (Continued)

infected samples). Infection was allowed to proceed for 1 hour to allow for virus entry, after which time media containing virus inoculum was replaced with dimethylsulfoxide (DMSO)- or AGK2-treated media, and the infection time course was initiated at 0 HPI. (B) Quantification of supernatant virus titers collected from cells treated with DMSO (vehicle control), 2.5- or 5.0-µM AGK2 at 120 and 144 HPI with either AD169 HCMV or TB40/E HCMV. (C) Confocal analysis of endogenous SIRT2 localization in uninfected cells and at 24, 48, 72, and 96 HPI with AD169 HCMV. Infection was verified by antibody-based detection of viral protein IE1 (24 and 48 HPI) or pUL99 (72 and 96 HPI). Nuclei were visualized by 4′,6-diamidino-2-phenylindole (DAPI) staining. Scale bars, 50 µM. (D) Diagram of the 5-day HCMV replication cycle, with numbers corresponding to the replication cycle stages analyzed in panels E–H. (E) Assessment of HCMV entry into host cells following DMSO or AGK2 treatment, analyzed using qPCR-based quantification of intracellular viral genomes at 0 HPI with AD169 HCMV. (F) Virus genomes produced at 48 or 72 HPI with AD169 HCMV in DMSO- or AGK2-treated cells, quantified by qPCR. (G) Titer of cell-associated HCMV collected from DMSO- or AGK2-treated cells at 72 HPI with AD169 HCMV. (H) Particle-to-infectious unit ratio of virus released by 120 HPI with AD169 HCMV from DMSO- or AGK2-treated cells. (I) Quantification of supernatant virus titers collected at 120 HPI with AD169 HCMV. For this assay, cells were not pre-treated with AGK2 prior to infection but were instead treated with either DMSO or AGK2 at 0, 24, 48, or 72 HPI. $N = 3$ biological replicates for panels B and E–I. Significance was determined by analysis of variance for panel B and Student's $t$-test for panels E–I. *$P \leq 0.05$, **$P \leq 0.01$, ***$P \leq 0.001$, ****$P \leq 0.0001$. Error bars indicate standard deviation. GO, gene ontology; HPI, hours post-infection; IP, immunoaffinity purification; LC-MS/MS, liquid chromatography tandem mass spectrometry; MS, mass spectrometry; n.s., not significant; qPCR, quantitative PCR.

in multiple subcellular locations. Therefore, given its diverse localization throughout infection, SIRT2 is poised to regulate different stages of the HCMV replication cycle.

To determine when SIRT2 acts during infection, we analyzed the ability of HCMV to proceed through each stage of the viral replication cycle following treatment with AGK2. For all assays, we used 2.5 µM AGK2, the lowest tested concentration that resulted in limited cytotoxicity and decreased HCMV replication for both viral strains tested (Fig. 1; Fig. S1A). We first measured viral entry and genome replication using quantitative polymerase chain reaction (qPCR). HCMV was able to enter AGK2-treated cells (Fig. 1E), but the treatment reduced viral genome replication at 48 and 72 HPI (Fig. 1F). By quantifying cell-associated infectious virus particles as a parameter for viral assembly, we found reduced titer, indicating that AGK2 treatment impacts assembly of infectious progeny virions (Fig. 1G). We additionally observed that AGK2 treatment resulted in a nearly fourfold increase in the particle:infectious unit (IU) ratio of virus released from infected cells (Fig. 1H). This signifies that virions produced from AGK2-treated cells are less infectious, as there is a higher ratio of viral particles released compared to infectious units. These results demonstrate that the defect in HCMV replication elicited by SIRT2 inhibition occurs after viral entry but prior to viral genome replication. These analyses were performed following pre-treatment of cells with AGK2, 12 hours prior to HCMV infection (Fig. 1A). To further define the temporality of the AGK2 phenotype, we next performed a dosing analysis by infecting fibroblasts that were not pre-treated with AGK2 (Fig. 1I). In this experiment, AGK2 was added to cell culture media post-infection at time points spanning the HCMV replication cycle (0, 24, 48, and 72 HPI). We found that addition of AGK2 at 0 HPI, immediately following infection with HCMV, decreased HCMV production (Fig. 1I). AGK2 addition at all other time points did not impact HCMV replication, demonstrating that inhibition of SIRT2 restricts HCMV replication prior to 24 HPI.

## SIRT2 interacts with known substrates and regulators of gene expression, metabolism, and cell cycle progression during early stages of HCMV replication

Our finding that AGK2 treatment elicits an anti-viral response suggests a function for SIRT2 deacetylase activity in support of HCMV replication. SIRT2 is known to be multi-functional, being positioned to potentially regulate diverse cellular processes that are important during early stages of HCMV infection, including gene expression, cellular metabolism, cytoskeletal organization, and cell cycle progression. Therefore, to investigate which of these cellular processes SIRT2 regulates during infection, we analyzed the temporal protein interactions of endogenous SIRT2. Immunoaffinity purification (IP) of SIRT2 was performed from uninfected cells and HCMV-infected cells at 0, 6, 12, and 24 HPI (Fig. 2A). These infection time points were chosen based on

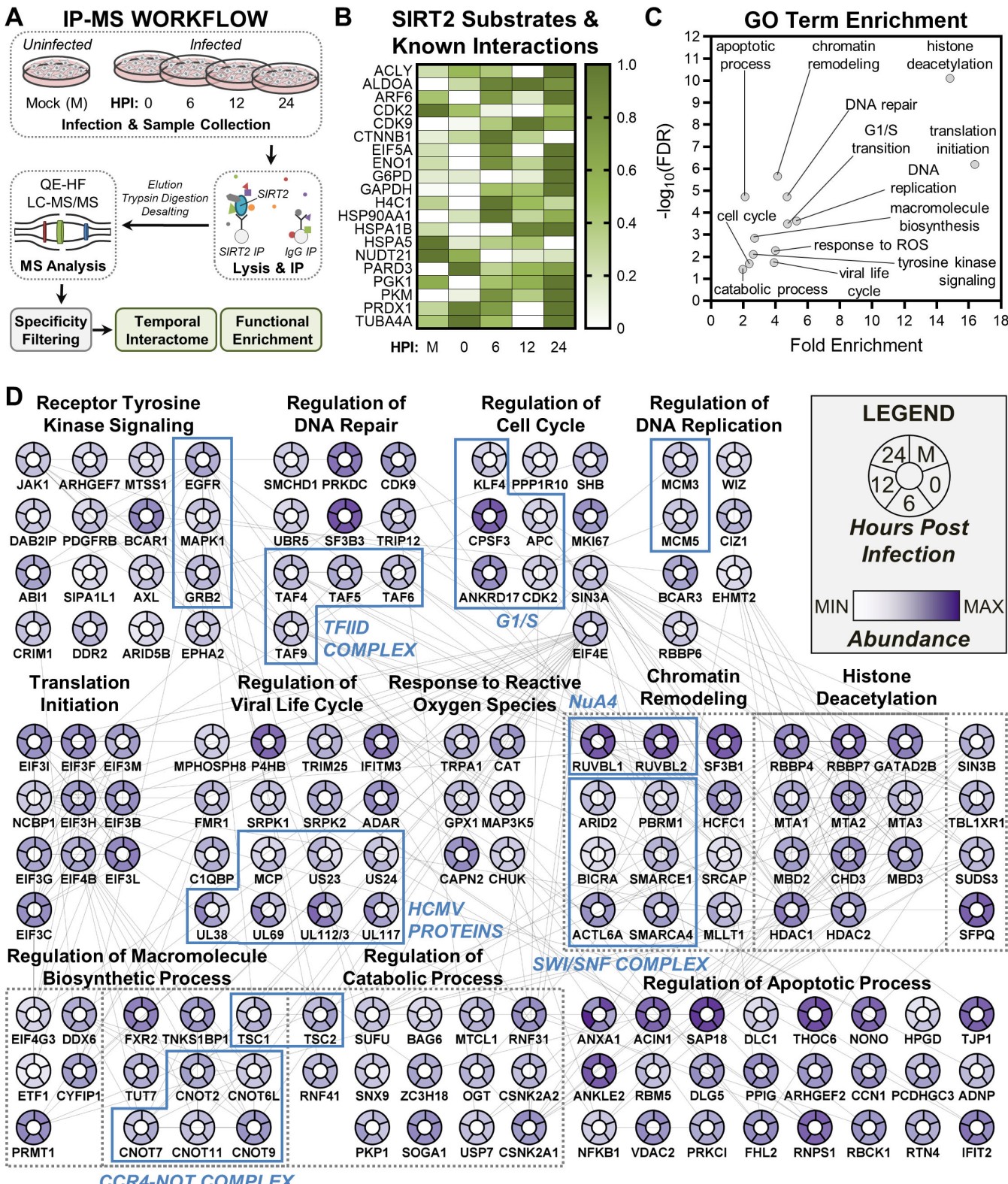

**FIG 2** IP-MS analysis identifies SIRT2 interactions with known substrates and proteins belonging to diverse cellular processes. (A) Diagram of the IP-MS workflow utilized for investigating the temporal protein interactions of endogenous SIRT2 during AD169 HCMV infection. (B) Heatmap showing the relative abundance of SIRT2 substrates or previously identified SIRT2-interacting proteins that were also detected in our IP-MS data set. Min-max normalization was performed to visualize at what time point the most abundant interaction occurs for each protein with SIRT2. (C) Gene ontology enrichment analysis of the 380 proteins that

**FIG 2** (Continued)

passed the specificity criteria (a statistically significant abundance fold change greater than 4) to be considered high-confidence SIRT2 interacting proteins. (D) Functional network of SIRT2 protein interactions. All proteins shown in the network passed the specificity criteria (a statistically significant abundance fold change greater than 4) for at least one analyzed time point. Network nodes indicate the protein abundance at each time point. Lines connecting nodes indicate known or predicted protein-protein interactions retrieved from the STRING database (79). Proteins are grouped by GO term, with dashed gray boxes grouping proteins that belong to multiple GO terms. Proteins boxed in blue are discussed in the Results. $N = 3$ biological replicates.

our finding that SIRT2 deacetylase activity is important early in infection (Fig. 1). The efficiency of SIRT2 isolation was confirmed by Western blot analysis (Fig. S2A). Mass spectrometry (MS) analysis was used to identify the proteins co-isolated with SIRT2, and the reproducibility of the biological replicates ($N = 3$) was verified by abundance distribution, coefficient of variation (CV%) frequency, and principal component analysis (Fig. S2B through D).

Twenty known SIRT2 substrates or previously recognized protein interactors were detected in our data set, including α-tubulin, metabolic enzymes, and cyclin-dependent kinases (Fig. 2B). HCMV capsids require the microtubule network for trafficking from the plasma membrane to the nucleus (80), and we found SIRT2 to have a maximal interaction with α-tubulin (TUBA4A) at 0 HPI, immediately following HCMV entry into cells. Not surprisingly, this interaction is retained, as prior work has shown the relevance of microtubule acetylation in spatial coordination between the nucleus and the viral assembly compartment (23).

Of the 20 known SIRT2 substrates or protein interactors detected in our data set, 7 function in central carbon metabolism. ALDOA, ENO1, GAPDH, PGK1, and PKM are glycolytic enzymes, while ACLY functions in fatty acid synthesis and G6PD is the rate-limiting enzyme of the pentose phosphate pathway. HCMV dynamically alters central carbon metabolism during its replication cycle to drive energy and biomolecule production (81–83). Increased glucose uptake and oxidation are induced by 24 HPI (76, 84), which coincides with when we see SIRT2 to exhibit maximal interaction with all seven metabolic enzymes (Fig. 2B).

Two cyclin-dependent kinases (CDK2 and CDK9) were also detected in our SIRT2 IP-MS data set (Fig. 2B). Humans possess 20 cyclin-dependent kinase (CDK) proteins, which are evolutionarily conserved enzymes that act as key regulators of cell cycle progression and transcriptional regulation (85). SIRT2 has been demonstrated to be a CDK2 substrate, with CDK2-mediated phosphorylation at S331 inhibiting SIRT2 catalytic activity (86). In contrast, CDK9 is a substrate of SIRT2. SIRT2 deacetylates CDK9 at K48 in response to stress, which stimulates CDK9 activity and promotes recovery from replication arrest (87). CDK proteins have been found to play roles during HCMV infection, with the maintenance or regulation of their activities contributing to alterations in cell cycle progression needed for HCMV replication (75, 88, 89). We found the SIRT2-CDK2 interaction to be most abundant in uninfected cells, whereas the SIRT2-CDK9 interaction was enhanced at 12 HPI.

After analyzing our data set for known SIRT2 substrates and interacting proteins, we next looked for high-confidence protein interactions using *P* value and abundance-based filtering (Fig. S2E). A stringent cutoff of statistically significant fourfold enrichment in SIRT2 IPs relative to IgG controls at any analyzed time point was used for identifying interactions. In total, 380 proteins passed this specificity threshold. The majority of the interacting proteins were annotated as nuclear or possessing dual localization to the nucleus and cytoplasm (Fig. S2F). Depending on the infection time point, 15%–18% of SIRT2 interacting proteins were designated as cytoplasmic and 2.8%–3.5% as mitochondrial. These results suggest that SIRT2 primarily regulates nuclear and cytoplasmic rather than mitochondrial processes during HCMV infection.

Gene ontology (GO) enrichment analysis of the 380 interacting proteins implicated SIRT2 in broadly regulating biological processes related to gene expression and cell division (Fig. 2C). Indeed, the resulting functional interaction network (Fig. 2D) showed enriched SIRT2 interactions with components of the epidermal growth factor signaling

pathway, including epidermal growth factor receptor (EGFR), mitogen-activated protein kinase 1, and growth factor receptor-bound protein 2 (GRB2). SIRT2 has been shown to regulate EGFR at the transcriptional level (90), and it remains to be seen whether this regulation extends at the protein level.

Another evident interaction cluster was that with chromatin remodeling proteins, histone acetyltransferase complexes, and related histone deacetylase proteins, including HDAC1 and HDAC2 (Fig. 2D). SIRT2 has abundant interactions with RUVBL1 and RUVBL2, two components of the NuA4 histone acetyltransferase complex (91), at all analyzed time points. Among interactions were six components of the SWI/SNF chromatin remodeling complex (ARID2, PBRM1, SMARCA4, SMARCE1, BICRA, and ACTL6A) (92), further implicating SIRT2 in regulating histone acetylation status and chromatin organization in both uninfected and infected fibroblasts. SIRT2 also formed associations with the CCR4-NOT complex (5 out of 10 subunits) and TFIID complex (four subunits), two supramolecular assemblies involved in gene expression (93, 94). SIRT2 has not yet been implicated in regulating the activities of these complexes.

We also detected numerous interactions between SIRT2 and cell cycle proteins (Fig. 2D). SIRT2 is a known regulator of cell cycle progression, having previously been shown to regulate the G2-M transition and mitotic checkpoint (57, 78, 95). Interestingly, rather than G2/M proteins, we found an enrichment of SIRT2 interactions with proteins involved in the G1-S transition. Of note were SIRT2 associations with CDK2 and ANKRD17. Following binding and activation by cyclin E, CDK2 drives cell cycle progression through the G1-S transition by phosphorylating its substrates. One such substrate of the cyclin E/CDK2 complex is ANKRD17 (96). We detected an abundant SIRT2-ANKRD17 interaction at all analyzed time points, further implicating SIRT2 in regulating CDK2 function.

In addition to host proteins, we detected SIRT2 interactions with seven HCMV proteins: MCP, pUS23, pUS24, pUL38, pUL69, pUL112/113, and pUL117. SIRT2 was previously identified to interact with the HCMV protein pUL133 (97, 98). The absence of this association in our data set may be linked to the temporality of this interaction. Our IP-MS experiment focused on early time points of infection, with our latest time point being 24 HPI, while the SIRT2-UL133 interaction was previously reported at 60 HPI. The interactions between SIRT2 and pUL38 and pUL117 are interesting, as these viral proteins have been shown to regulate the activities of host proteins that we also detect as high-confidence SIRT2 interacting proteins. pUL117 inhibits host DNA replication by targeting the mini-chromosome maintenance (MCM) complex (99), and we identified two components of the MCM complex, MCM3 and MCM5, to interact with SIRT2. We also observed an interaction between SIRT2 and TSC1 and TSC2, host factors targeted by pUL38 to regulate mTORC1 activity and protein synthesis (100). Altogether, our analysis of temporal protein interactions suggests that SIRT2 is poised to broadly regulate cellular processes related to gene expression and cell cycle progression during infection.

## SIRT2 modulates the acetylation of cell cycle proteins during infection

Gene expression and cell cycle progression are two tightly orchestrated and regulated biological processes that involve numerous protein factors. To pinpoint which proteins belonging to these processes SIRT2 regulates during HCMV infection, we sought to characterize its deacetylase activity substrates. Given that there is still limited information about the global effect of SIRT2 on the cellular acetylome, especially in a primary cell type, we investigated changes in the protein acetylome and proteome following AGK2 treatment and HCMV infection (Fig. 3A). Cells treated with AGK2 or DMSO were infected with HCMV (0, 12, or 24 HPI), and acetylated peptides were enriched and subjected to MS analysis. Acetylome and proteome data set quality were assessed by protein abundance distribution, linear regression modeling, CV% frequency distribution, and principal component analysis (Fig. S3).

Our acetylome data set consisted of 5,452 unique acetylated peptides belonging to 2,002 proteins (Fig. 3B). Of these, 393 acetylated peptides, belonging to 338 proteins, displayed increased abundance with AGK2 treatment or were only detected in the

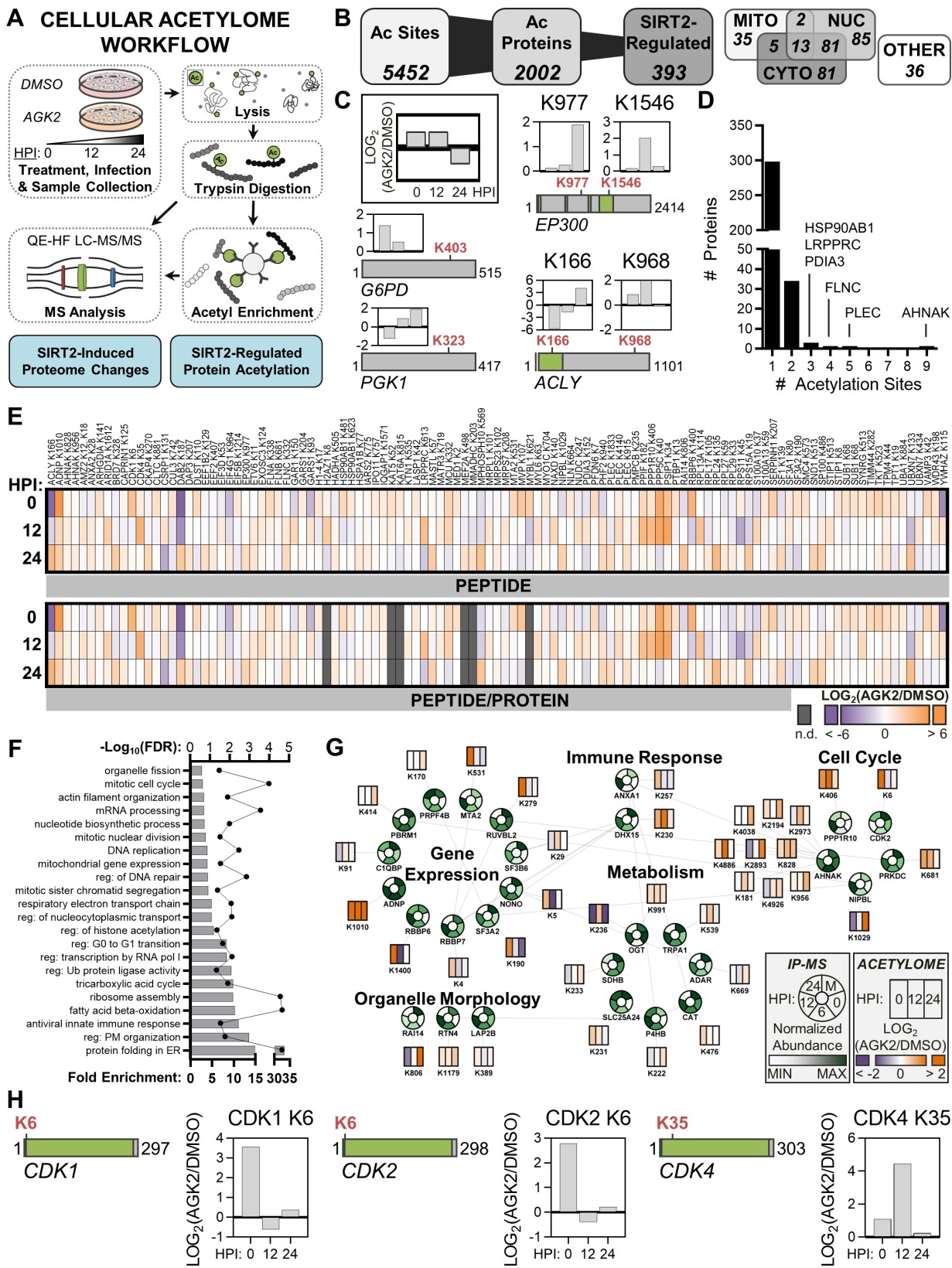

FIG 3 Acetylome analysis reveals altered acetylation status of cell cycle proteins following inhibition of SIRT2 with AGK2 treatment. (A) Acetylome workflow for investigating the impact of SIRT2 on protein abundance and acetylation status during AD169 HCMV infection. (B) Diagram depicting the total number of unique acetylated peptides identified in our data set with the proportion that were increased in abundance following SIRT2 inhibition. Subcellular localization

**FIG 3** (Continued)

analysis was performed with the parent proteins of the 393 differentially regulated acetylated peptides using GO and UniProt databases. (C) Diagrams of known SIRT2 substrates with temporally altered lysine acetylation following SIRT2 inhibition. Protein length, acetylated lysine location, and functional domains (green) are shown. See also Fig. S4D. Bar graphs depict the temporal abundance of each acetylated peptide at 0, 12, and 24 HPI with AGK2 treatment relative to DMSO control. (D) The number of acetylation sites per protein that increased in abundance with SIRT2 inhibition. (E) Heatmap of acetylated peptides that increased in abundance at least twofold with AGK2 treatment relative to DMSO control. The top heatmap shows acetylated peptide abundances. The bottom heatmap shows the protein-normalized acetylated peptide abundances. (F) Gene ontology enrichment analysis of all proteins containing acetylation sites that were increased in abundance with AGK2. The dotted line provides false discovery rate (FDR), and the bars correspond with fold enrichment. (G) Proteins that were identified as high-confidence SIRT2 interacting proteins in our IP-MS analysis, which also contain an acetylated lysine that increased in abundance with SIRT2 inhibition. Donut charts depict the min-max normalized temporal abundance of each protein in the IP-MS data set. Rectangular heatmaps depict the temporal fold change in abundance of acetylated peptides belonging to each protein. (H) Diagrams of cyclin-dependent kinase proteins with temporally altered lysine acetylation driven by SIRT2 inhibition. Protein length, acetylated lysine location, and functional domains (green) are shown. Bar graphs depict the temporal abundance of each acetylated peptide at 0, 12, and 24 HPI with AGK2 treatment relative to control. $N$ = 2 biological replicates.

treated samples (not controls), suggesting that these sites represent potential substrates of SIRT2 (Fig. S4A and B). Fitting with the known subcellular localization pattern of SIRT2, we annotated most of these putative substrates to be nuclear, cytoplasmic, or dually localized to both compartments (Fig. 3B). Indeed, among these were G6PD, EP300, ACLY, and PGK1, known substrates of SIRT2 (Fig. 3; Fig. S4D) (62, 101–103). Notably, G6PD, ACLY, and PGK1 were also detected in our IP-MS analysis of SIRT2 interactions during infection (Fig. 2B). Deacetylation of G6PD K403 by SIRT2 increases metabolic flux through the pentose phosphate pathway (102). We detected a slight increase in K403 acetylation at 0 HPI upon SIRT2 inhibition. SIRT2 is also known to deacetylate K1546 of the histone deacetylase EP300 (101), which restores EP300 binding to the transcription preinitiation complex. We observed a twofold increase in acetylation of this site at 12 HPI with AGK2 treatment. While SIRT2 has previously been shown to deacetylate ACLY K540, K546, and K554 (104), SIRT2 regulation of K166 and K968, the sites identified in our data set, has not been demonstrated. Notably, K166 lies within the ATP-grasp domain of ACLY, which is required for catalysis (105). PGK1 has also been previously reported as a SIRT2 substrate (106); however, the specific lysine residues targeted by SIRT2 have not been reported. Here, we detect twofold increased acetylation of PGK1 K323 with SIRT2 inhibition, suggesting this as a site regulated by SIRT2.

Several putative substrates contained multiple acetylated sites modulated by SIRT2 inhibition (Fig. 3D). Of these, HSP90AB1, LRPPRC, and PDIA3, each with three acetylation sites, function in protein folding, RNA metabolism, and disulfide bond formation, respectively. FLNC and PLEC, which both function in cytoskeleton organization, also possessed multiple regulated acetylated sites. Most regulated sites were observed on AHNAK, a protein that functions in cell cycle and cell migration.

Sequence motif analysis of acetylated peptides that increased upon AGK2 treatment showed an enrichment for acidic amino acids immediately N-terminal of the acetylated lysine and for lysine residues at its C-terminus (Fig. S4C) (107). This finding agrees with a recent acetylome analysis conducted following SIRT2 overexpression or knockdown in colorectal cancer cells, which also found an enrichment for lysine residues C-terminal of the acetylated lysine regulated by SIRT2 (65). Although this observation may be partly driven by the workflow (i.e., trypsin digestion), the enrichment for lysine at six C-terminal amino acid positions suggests that SIRT2 might target acetylation sites within regulatory domains (e.g., modifiable by multiple PTMs or enzymes that target K residues, NLS, disordered regions).

We next focused on acetylated peptides that were significantly increased in abundance, at least twofold, with AGK2 treatment. A total of 111 peptides passed the twofold change abundance threshold for at least one analyzed infection time point (Fig. 3E). Normalization of acetylated peptide abundance to protein abundance did not alter the overall abundance trends, demonstrating that the differential acetylation levels are due to regulation at the PTM level rather than at the protein level. However, there were significant changes in the cellular proteome with AGK2 treatment relative to DMSO

treatment, supporting that SIRT2 is a multi-functional protein that can profoundly alter the cellular state (Fig. S5). Functional enrichment analysis revealed that SIRT2 inhibition alters the acetylation status of proteins functioning in diverse biological processes, including metabolism, cell cycle, gene expression, and organelle structure-function relationships (Fig. 3F).

To identify likely high-confidence substrates, we integrated the acetylome and SIRT2 interactome data sets, focusing on proteins present in both data sets. Twenty-eight of the putative substrates (from the acetylome) were also identified as high-confidence SIRT2 protein interactions (>4 fold enrichment versus control) (Fig. 3G). RUVBL2, ADNP, SF3A2, CDK2, PRKDC, and NONO were particularly interesting as the increase in acetylation following SIRT2 inhibition temporally aligned with the time point of maximal interaction with SIRT2 during infection. RUVBL2 possesses DNA helicase activity and is a component of the NuA4 histone acetyltransferase complex, which was also enriched in the SIRT2 interactome (Fig. 2D). ADNP is also involved in transcriptional regulation, while SF3A2 functions in mRNA splicing. CDK2, PRKDC, and NONO function in the DNA damage response and cell cycle progression. The function of many of these proteins in cell cycle regulation hints toward a function for SIRT2 in regulating this process during early stages of HCMV replication.

The integration of the interactome and acetylome data sets highlighted several CDK proteins as likely SIRT2 substrates, including CDK1, CDK2, and CDK4. To assess functional consequences of increased acetylation, we investigated whether differentially regulated acetylated lysine residues resided in known functional domains of these CDK proteins (Figure 3; Fig. S4D). CDK1, CDK2, and CDK4 all function in regulating cell cycle progression, specifically the G1-S transition. We found the K6 residue of both CDK1 and CDK2 to be significantly increased in acetylation with AGK2 treatment. CDK1 K6 acetylation increased 3.6-fold at 0 HPI and CDK2 K6 acetylation increased 2.8-fold at the same time point. K6 falls within the catalytic protein kinase domain of CDK1 and CDK2, suggesting that acetylation of this site could regulate kinase activity. Acetylation of these sites has previously been shown; however, their regulation and functional consequences are unknown (2). Acetylation of CDK4 K35 was increased 4.5-fold at 12 HPI with AGK2 treatment. K35 is the catalytic lysine of CDK4 (108) and has not been previously shown to be acetylated. However, acetylation of the catalytic lysine residue of CDK1, CDK2, CDK5, and CDK9 has been shown to inhibit kinase activity (9, 87, 109–112), suggesting a conserved mechanism for CDK regulation.

## SIRT2 inhibition and CDK2 (K6) acetylation promote the G1/S cell cycle transition

Given the enrichment of cell cycle proteins in our interactome and acetylome data sets, we hypothesized that SIRT2 functions to regulate cell cycle progression during HCMV replication. To test this, we performed a flow cytometry-based cell cycle profiling experiment (Fig. 4A). Mitotic cells were identified by the presence of phosphorylated histone H3 S10 (pH3-Ser10). To distinguish the S phase, cells were treated with the nucleotide analog EdU, which is incorporated into newly synthesized DNA. Cells in G1 and G2 were differentiated by DAPI staining for DNA content; cells in G2 possess double the DNA content of cells in G1. We performed this analysis following AGK2 treatment in uninfected cells and HCMV-infected cells at 0, 6, 12, and 24 HPI. Upon SIRT2 inhibition, cell percentages were decreased in the G1 phase and increased in the S phase (Fig. 4B and C). This phenotype was apparent in uninfected cells and the first three time points of infection (0, 6, and 12 HPI). By 24 HPI, this phenotype was lost. Additionally, AGK2 treatment did not impact the percentage of cells in the G2 or M phase relative to DMSO treatment. Our finding that AGK2 treatment impacts cell proliferation is further supported by an increase in cell count by 144 HPI in AGK2-treated cells relative to DMSO-treated cells (Fig. S6A).

Cell cycle progression is regulated by the inhibition and activation of cyclin-dependent protein kinases (Fig. 4D) (113). CDK4 and CDK6 interact with cyclin D1 and regulate

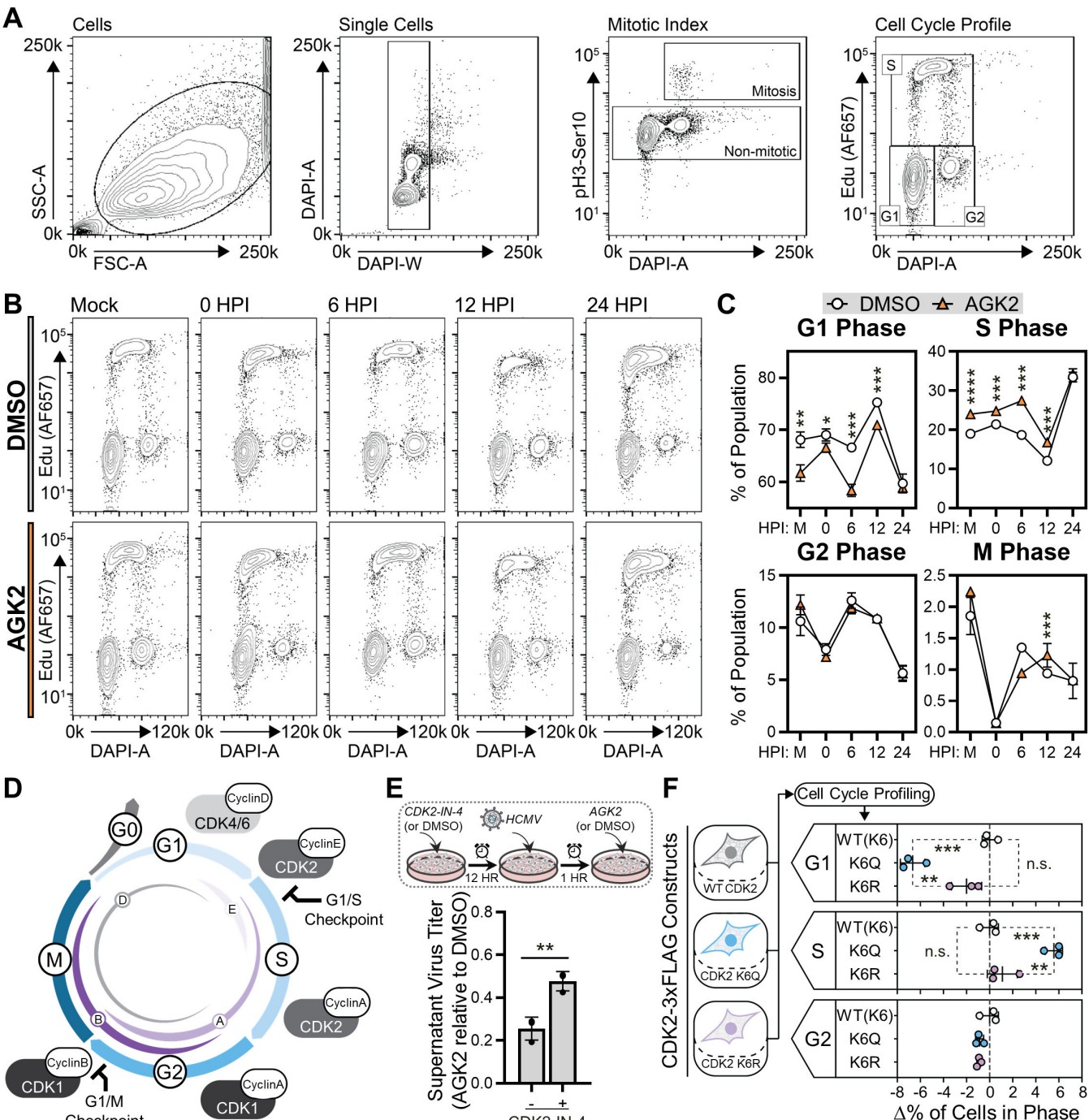

**FIG 4** SIRT2 inhibition and CDK2(K6) acetylation drive cell cycle progression from G1 to S phase. (A) Illustrative contour maps depicting the gating parameters utilized for flow cytometry-based cell cycle profiling experiments. Single cells were isolated by gating based on FSC, SSC, and DAPI profiles. Mitotic cells were isolated based on the presence of phosphorylated histone H3 S10 (pH3-S10) and non-mitotic cells were further gated into G1, S, or G2 phase populations. Cells in S phase were gated based on EdU staining, as EdU is incorporated into newly synthesized DNA. Cells in G1 and G2 were gated based on DAPI profiles, as cells in G2 possess double the DNA content of cells in G1. (B) Representative contour maps for both treatment conditions (DMSO or AGK2) showing the proportion of cells in G1, S, or G2 phase. The flow cytometry-based cell cycle profiling experiment was performed for uninfected cells (mock) and infected cells at 0, 6, 12, and 24 HPI with AD169 HCMV. (C) Quantification of the data shown in panel B across three biological replicates. The average percentage of cells in each cell cycle stage is shown for each analyzed infection time point and treatment condition. (D) Cell cycle diagram depicting the cell cycle stages (G0, G1, S, G2, and M) and the key cyclin and CDK proteins that regulate progression through and between stages. (E) Quantification of supernatant virus titer at 144 HPI following treatment with a CDK2 inhibitor (CDK2-IN-4) and AGK2. Cells were incubated with (+) or without (−) CDK2-IN-4 for 12 hours prior to infection with AD169 HCMV. (Continued on next page)

**FIG 4** (Continued)

AGK2 or DMSO (vehicle control) were added to the cells at 0 HPI, following removal of HCMV inoculated cell media. Virus titer was calculated as AGK2-treated samples relative to DMSO-treated samples for each CDK2-IN-4 treatment condition. (F) Quantification of the change in the percentage of cells existing in G1, S, or G2, driven by stable expression of acetyl mimic CDK2(K6Q) or lysine charge mimic CDK2(K6R) constructs relative to WT CDK2(K6) expression. $N = 3$ biological replicates. Significance was determined by Student's *t*-test for panels C and E and analysis of variance for panel F. *$P \leq 0.05$, **$P \leq 0.01$, ***$P \leq 0.001$, ****$P \leq 0.0001$. Error bars indicate standard deviation. FSC, forward scatter; SSC, side scatter.

G1-phase progression; CDK2 first interacts with cyclin E to initiate S-phase entry and then transitions to interact with cyclin A during processive DNA replication. Finally, CDK1 associates with cyclin A and then cyclin B1 to coordinate G2-phase duration and M-phase entry. Hence, CDK2 is the major regulator of cell cycle progression from G1 to S phases (Fig. 4D) (114). Based on our cell cycle profiling results, our identification of a SIRT2-CDK2 interaction at early time points of infection, and increased CDK2 K6 acetylation following SIRT2 inhibition (Fig. 3G), we therefore focused our follow-up investigations on CDK2. We asked whether the increase in CDK2 acetylation is linked to a disruption in the SIRT2-CDK2 interaction upon SIRT2 inhibition. Indeed, using an IP-MS analysis, we observed that AGK2 treatment resulted in a slight decrease in the levels of CDK2 co-isolated with SIRT2 (Fig. S6B). Next, we investigated whether SIRT2 functions through the regulation of CDK2 activity during HCMV infection. To test this, we inhibited CDK2 activity prior to HCMV infection using the pharmacological inhibitor CDK2-IN-4 (115). Cells either pre-treated or not treated with CDK2-IN-4 were infected with HCMV and then subjected to AGK2 treatment at 0 HPI (Fig. S6C). Compared to those not exposed to CDK2 inhibition, pre-treatment with CDK2-IN-4 resulted in a 50% increase in virus titer in AGK2-treated cells (Fig. 4E). These results indicate that SIRT2-mediated regulation of CDK2 activity is at least in part responsible for the effect on virus titer.

We next sought to confirm the specific involvement of CDK2 K6 acetylation status in our observed cell cycle phenotype upon SIRT2 inhibition. MRC5 fibroblasts stably expressing either wild-type (WT) CDK2-3×FLAG, an acetyl mimic CDK2(K6Q)–3×FLAG, or a lysine charge mimic CDK2(K6R)–3×FLAG construct were generated (Fig. S6D). Flow cytometry-based cell cycle profiling of these cell lines recapitulated our findings with SIRT2 inhibition (Fig. 4F). Expression of the acetyl mimic CDK2(K6Q) construct increased the proportion of cells in the S phase and decreased the proportion of cells in G1 relative to expression of WT CDK2. Expression of the lysine charge mimic CDK2(K6R) construct had no significant impact on the proportion of cells in G1, S, or G2 phase relative to WT CDK2 expression. Altogether, these findings demonstrate that SIRT2 inhibition via AGK2 treatment results in increased acetylation of CDK2 K6, which induces the cell cycle G1- to S-phase transition.

## SIRT2 inhibition results in delayed HCMV replication kinetics

Like many nuclear-replicating DNA viruses, HCMV infects quiescent cells and drives entrance into G1 phase to initiate processes supporting genome replication, while inhibiting G1-S transition to avoid competition with host DNA replication (66, 67, 116). HCMV additionally possesses mechanisms to sense if an infected cell is in the correct cell cycle state for viral replication and, if not, will stall replication until the cell re-enters G1 (117–119). Therefore, the increased proportion of cells in S phase driven by SIRT2 inhibition and increased CDK2 K6 acetylation would stall HCMV replication, as cells would need to cycle through mitosis and return to G1 for virus replication to proceed. Therefore, we next tested the kinetics of HCMV replication following SIRT2 inhibition by quantifying viral protein abundances throughout infection. For this, we applied our TRUSTED assay that allows the simultaneous quantification of all temporal classes of viral proteins via targeted mass spectrometry (Fig. 5A) (120). We found SIRT2 inhibition to broadly decrease the temporal abundance of HCMV immediate-early (IE), delayed early, leaky late, and late proteins (Fig. 5A).

By performing a Euclidean distance analysis of the viral proteome at all analyzed infection time points, we observed a temporal delay in HCMV protein levels upon SIRT2

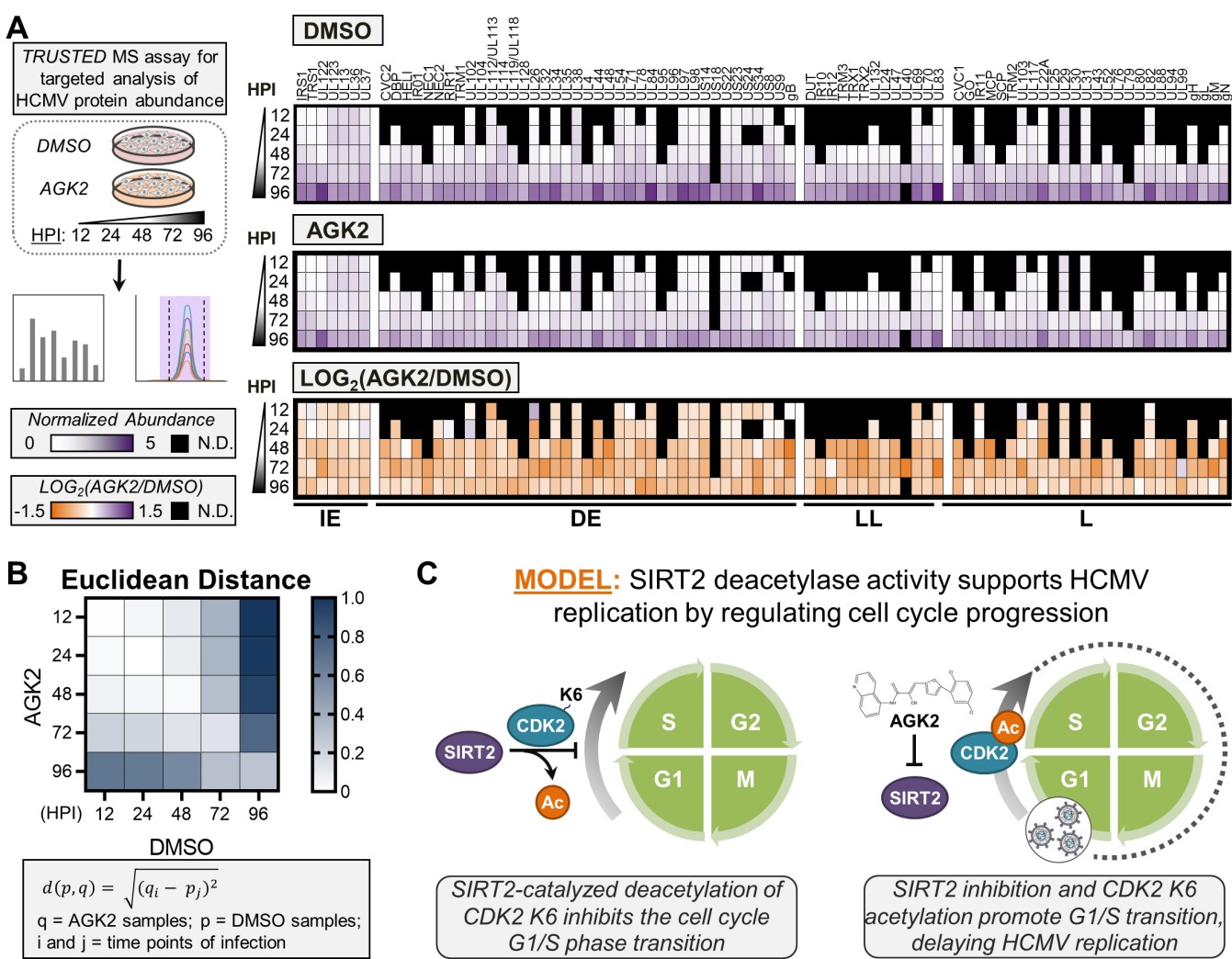

**FIG 5** SIRT2 inhibition delays HCMV replication kinetics. (A) Our TRUSTED targeted mass spectrometry assay (120) was utilized for detection and quantification of HCMV protein abundances throughout infection with AD169 HCMV following DMSO or AGK2 treatment. Heatmaps depict the abundance of HCMV proteins throughout infection following treatment with DMSO (top heatmap) or AGK2 (center heatmap). The bottom heatmap depicts the fold change in abundance (AGK2 relative to DMSO treatment) for each viral protein across analyzed infection time points. Proteins are grouped by their temporal class: IE, DE, LL, and L. $N$ = 3 biological replicates. (B) Euclidean distance analysis showing the impact of SIRT2 inhibition on the temporality of HCMV protein abundance profiles. (C) Proposed model for the function of SIRT2 deacetylase activity in promoting HCMV replication. DE, delayed early; IE, immediate-early; L, late; LL, leaky late;N.D., not detected.

inhibition (Fig. 5B). This is most apparent at later infection time points, when HCMV proteins are most abundant within the infected cell. For instance, at 48 HPI, the viral proteome of AGK2-treated cells was equivalent to that of 24 and 48 HPI DMSO-treated cells. Similarly, the viral proteome of 72-HPI AGK2-treated cells was equivalent to 48- and 72-HPI control cells, and AGK2-treated cells at 96 HPI were similar to 72- and 96-HPI DMSO-treated cells. The delay in HCMV replication kinetics following AGK2 treatment was supported by a multi-step growth curve analysis (Fig. S6E). At 5 days post-infection (DPI), AGK2 treatment resulted in the same previously observed decrease in virus titer relative to DMSO treatment. However, by 10 DPI, the virus titer for AGK2-treated cells recovered to the same level as for DMSO-treated cells.

These findings lead to a model for the mechanism underlying the pro-virus effect of SIRT2 deacetylase activity during HCMV infection (Fig. 5C). We propose that SIRT2 normally functions to inhibit cell cycle progression from G1 to S phase during HCMV infection, which promotes maintenance of cells in G1, the cell cycle stage that supports

HCMV replication. Furthermore, our results support that this is accomplished partly through the regulation of CDK2 K6 acetylation status. Inhibition of SIRT2 deacetylase activity results in increased acetylation of CDK2 K6, which drives the G1- to S-phase transition and delays HCMV replication.

## DISCUSSION

Here, we mechanistically define the function of SIRT2 deacetylase activity in supporting HCMV replication. Our IP-MS and acetylome analyses revealed SIRT2 interactions with key cell cycle regulators, including CDK2, and increased acetylation of these proteins upon SIRT2 inhibition. Flow cytometry-based cell cycle profiling supported these findings, demonstrating that SIRT2 functions to inhibit cell cycle progression from G1 to S phases. Inhibition of SIRT2 deacetylase activity promoted cell cycle progression into S phase, a non-productive cell cycle state for HCMV replication. This resulted in delayed initiation of HCMV replication, which was reflected in the altered kinetic profiles of HCMV protein abundances upon SIRT2 inhibition. Interestingly, AGK2 treatment has also been shown to inhibit the replication of hepatitis B virus (HBV) through an unknown mechanism (41). Similar to HCMV, HBV is a nuclear-replicating virus that dysregulates cell cycle progression to promote its replication (121, 122). It is possible that AGK2 inhibits HBV replication through a similar mechanism by opposing virus-induced cell cycle dysregulation.

In studying the provirus effect of SIRT2 during HCMV infection, we have also identified and characterized an SIRT2-regulated acetylation site on CDK2. We found CDK2 to interact with SIRT2 in uninfected cells and at early time points of HCMV infection. We also observed acetylation of CDK2 K6 to increase ~3-fold at 0 HPI with SIRT2 inhibition. K6 lies within the protein kinase domain of CDK2 and has been previously found to be an acetylated residue (2). However, the function of the acetylation and its regulation throughout the cell cycle had not yet been investigated. Based on our finding that expression of a CDK2 K6 acetyl mimic (CDK2 K6Q) increases the population of cells in S phase, we propose that CDK2 K6 acetylation is involved in activating CDK2. Following from this, deacetylation of CDK2 K6 by SIRT2 would inactivate CDK2 and stall the cell cycle in G1. Through an interaction with cyclin E, CDK2 regulates cell cycle progression from G1 to S phases (114). Intriguingly, SIRT2 is known to be a substrate of the cyclin E-CDK2 complex, with phosphorylation of SIRT2 S331 functioning to inhibit SIRT2 activity (86). CDK2 being itself regulated by SIRT2 presents an interesting feedback mechanism for regulating cell cycle progression.

Our recent analysis of temporal alterations in protein acetylation throughout HCMV infection revealed that acetylation is broadly regulated throughout viral replication (11). Similar to our prior finding that lamin B1 acetylation inhibits HCMV production by obstructing virus capsid nuclear egress (11), the pro-viral function of SIRT2 deacetylase activity provides another instance of protein acetylation functioning in host defense during HCMV infection. However, in addition to functioning as a protein deacetylase, SIRT2 also removes long-chain fatty-acyl, lipoyl, benzoyl, 4-oxononanoyl, and methacrylation groups (33, 45–53). The multiple enzymatic activities of SIRT2 likely explain why siRNA-mediated knockdown of SIRT2 promotes HCMV replication, while specific inhibition of SIRT2 deacetylase activity inhibits HCMV replication. This dichotomy suggests that the different SIRT2 enzymatic activities and substrates function in both an anti- and pro-viral fashions, and this is likely the case for other SIRTs as well. Therefore, when investigating the roles of SIRTs during viral infection, it could be necessary to use specific small molecule inhibitors or directed ablation of certain enzymatic activities to gain a better understanding of their functions.

Outside the context of viral infection, SIRT2 has been implicated in cancer progression. However, whether SIRT2 functions as an oncogene or a tumor suppressor has been heavily debated (123). It is likely that, similar to its function during viral infection, SIRT2 functions in both capacities, depending on the enzymatic activity and substrate. Given our finding that SIRT2 regulates the G1-S transition and inhibition of its deacetylase

activity drives cell cycle progression, it is possible that SIRT2 deacetylase activity serves a tumor-suppressive function. Altogether, our findings have significant ramifications for understanding the function of SIRT2 during viral infection and reflect a necessity for deconvoluting the various enzymatic activities of SIRTs and their functions in human health and disease.

## MATERIALS AND METHODS

### Cell lines and virus strains

MRC5 primary human fibroblasts (ATCC CCL-171, passage numbers 18–28) were used for all experiments. Cells were grown in complete growth medium (DMEM supplemented with 10% fetal bovine serum) under standard conditions (37°C and 5% $CO_2$). Cells were tested for mycoplasma using the MycoStrip (Invivogen) Mycoplasma Detection Kit and were authenticated by cell morphology and growth curve analyses.

Human cytomegalovirus infections were performed using AD169 or TB40/E virus strains. P0 virus was produced by transfecting fibroblasts with bacterial artificial chromosomes containing the viral genome, as described in references 124, 125. To generate P1 virus, which was used for all experiments, fibroblasts were infected with P0 virus, and the P1 virus produced was collected and concentrated by ultracentrifugation. Virus titer was determined by tissue culture infectious dose (TCID50).

### AGK2 treatment and infection conditions

AGK2 stock solution was generated by resuspending powdered high-performance liquid chromatography (HPLC)-grade AGK2 (Fisher Scientific) in DMSO. Unless otherwise stated, cells were treated 12 hours prior to infection with either DMSO (vehicle control) or 2.5-µM AGK2 in complete growth media. A final growth media DMSO concentration of 0.1% was used for all treatments.

For HCMV infections, confluent fibroblasts were infected at a multiplicity of infection (MOI) of 1 for 1 hour at 37°C with shaking every 10 min. Complete growth media not containing AGK2 or DMSO were used for all infections. Following the 1-hour incubation, complete growth media containing AGK2 or DMSO were added back to the cells, and the infection time course was initiated at 0 HPI. The same procedure was used for mock infected cells but without viral inoculum added to the media. Unless otherwise stated, the AD169 HCMV strain was used for all experiments.

### Determination of AGK2 cytotoxicity

For assessing AGK2 cytotoxicity in our cell culture model, cells were plated in 96-well plates and treated at 70% confluency with 0 (negative control wells), 2.5, or 5.0 µM AGK2 in biological triplicate. Cells were infected with HCMV at an MOI of 1, 12 hours after AGK2 treatment. Cells were collected at 24 and 144 HPI for cell death detection using the In Situ Cell Death Detection Kit, TMR Red (Sigma 12156792910). The kit protocol was followed as written. Briefly, cells were washed with phosphate buffered saline (PBS) and fixed for 1 hour with 4% paraformaldehyde in PBS. After fixation, cells were washed with PBS and blocked for 10 min with 3% $H_2O_2$ in water. Cells were then washed with PBS and permeabilized with 0.1% Triton X-100 in 0.1% sodium citrate for 2 min at 4°C. After permeabilization, a positive control for cell death was generated by incubating one positive control well for 10 min with 10 U DNase I in 50 mM Tris-HCl (pH 8.0), 10 mM $MgCl_2$, 1-mg/mL bovine serum albumin (BSA). All wells were then washed with PBS and labeled with terminal deoxynucleotidyl transferase dUTP nick end labeling (TUNEL) reaction mixture. Negative control wells were labeled with TUNEL label solution not containing enzyme solution. Cells were protected from light and labeled for 1 hour at 37°C. Following TUNEL reaction labeling, cells were incubated for 10 min with 1:1,000 DAPI in PBS to stain nuclei for cell count analysis. Cells were washed and imaged using

the Operetta imaging system (PerkinElmer) with excitation wavelength of 520–560 nm and detection wavelength of 570–620 nm.

## Western blot analysis

Cells were collected for Western blot analysis by scraping into cold PBS and centrifuging. Cell pellets were washed twice with cold PBS, spiked with 100× HALT protease and phosphatase inhibitor (78438, ThermoFisher), snap frozen in liquid nitrogen, and stored at −80°C. For analysis of tubulin and histone H3 acetylation levels following AGK2 treatment, frozen cell pellets were thawed on ice and lysed in radioimmunoprecipitation assay buffer, pH 7.4 (Boston BioProducts, BP-115). For analysis of SIRT2 abundance throughout infection and confirmation of CDK2 construct expression, cells were lysed in 100-mM Tris (pH 8.0), 4% SDS, 1-mM EDTA.

Protein concentration was determined by Pierce bicinchoninic acid (BCA) assay (ThermoFisher) and 20 µg of protein was aliquoted per sample and reduced with 2-mercaptoethanol. For assessment of SIRT2 IP-MS isolation efficiency, equivalent volumetric ratios of IP input, solubilized pellet, IP flow-through, wash, elution 1, and elution 2 samples were collected during the IP-MS experimental procedure and reduced with 2-mercaptoethanol. Proteins were separated using SDS-PAGE and transferred to a polyvinylidene difluoride (PVDF) membrane overnight (16–18 hours) at 4°C.

To develop the Western blots, PVDF membranes were blocked in 5% milk in tris-buffered saline (TBS) for one hour and incubated with primary antibodies in TBS + 0.1% Tween 20 (TBST) containing 5% milk overnight at 4°C. For analysis of tubulin and histone H3 acetylation levels, PVDF membranes were washed with TBST and incubated with 1:10,000 anti-rabbit and anti-mouse light chain specific horseradish peroxidase (HRP)-conjugated (Jackson ImmunoResearch) secondary antibodies in TBST + 5% milk + 0.01% SDS for 1 hour prior to incubation with chemiluminescent solution (ECL, Millipore). Signal was detected using a UVP AutoChemi system (UVP, Upland). For analysis of SIRT2 abundance, SIRT2 IP-MS isolation efficiency, and confirmation of CDK2 construct expression, PVDF membranes were washed with TBST and incubated with 1:10,000 Alexa Fluor 680 goat anti-rabbit IgG (A27042, ThermoFisher) and Alexa Fluor Plus 800 goat anti-mouse IgG (A32730, ThermoFisher) antibodies in TBST + 5% milk + 0.01% SDS for 30 min. Developed membranes were washed with TBST, and fluorescent signal was detected using the Odyssey CLS system (Li-Cor). ImageStudio v.5.2 (Li-Cor) was used to quantify band intensity using densitometry.

The following antibody dilutions were used for primary incubation: 1:50,000 mouse anti-β-actin HRP (Abcam, ab49900); 1:1,000 rabbit anti-acK40 tubulin (Abcam, ab179484); 1:500 rabbit anti-SIRT2 antibody (LifeTechnologies, 19655-1-AP; Fig. S2A); 1:2,000 rabbit anti-SIRT2 antibody (Abcam, ab211033; Fig. S1F); 1:1,000 rabbit anti-GAPDH (Cell Signaling, CSIG-5174S); 1:10,000 mouse anti-FLAG (Sigma, F1804); 1:1,000 rabbit anti-histone H3 (Abcam, ab1791); and 1:1,000 mouse anti-histone H3K9ac (Thermo, 50-199-3271).

## Immunofluorescence microscopy

### Sample preparation and image acquisition

For analysis of SIRT2 localization during HCMV infection, cells were seeded on fibronectin-coated glass slides and infected at 100% confluency with HCMV at an MOI of 1. Uninfected cells and infected cells at 24, 48, 72, and 96 HPI were washed with PBS, fixed for 15 min with 4% paraformaldehyde (PFA) in PBS, and permeabilized for 10 min with 0.2% Triton X-100 in PBS (0.2% PBS-T). Following fixation and permeabilization, cells were rinsed with PBS and blocked with 5% goat serum and 2.5% human serum in 0.2% Triton X-100 for 30 min. Cells were incubated with 1:200 rabbit anti-SIRT2 (Abcam, ab67299) and either 1:40 mouse anti-IE1 (clone 1B12, gift from Thomas Shenk [126], 24- and 48-HPI samples) or 1:100 mouse anti-pUL99 (clone 10B4, gift from Thomas Shenk [127], 72- and 96-HPI samples) in block overnight at 4°C. Following primary incubation, cells were

washed with 0.2% PBS-T and incubated with 1:2,000 AlexaFluor 488 goat anti-rabbit and 1:2,000 AlexaFluor 555 goat anti-mouse secondary antibodies with 1:1,000 DAPI in block for 1 hour at room temperature. The cells were then washed with 0.2% PBS-T prior to mounting with Aqua-Poly/Mount medium (Polysciences, Inc., Warrington, PA). Immunofluorescence microscopy analysis of SIRT2 localization was conducted using a LEICA SP5 confocal microscope (Leica Microsystems Inc., Buffalo Grove, IL).

For analysis of tubulin acetylation levels following AGK2 treatment, cells were plated in glass-bottom culture dishes (MatTek Corporation) and treated with DMSO or 2.5 µM AGK2 at 70% confluency. Cells were washed with PBS, fixed for 15 min with 4% PFA, and permeabilized with ice-cold 100% MeOH for 15 min at −20℃. Following fixation and permeabilization, cells were rinsed with PBS and blocked with 10% goat serum in 0.1% Triton X-100 in PBS (0.1% PBS-T) for 30 min. Cells were incubated with 1:100 rabbit anti-acK40 tubulin (Abcam, ab179484) and 1:500 anti-tubulin (Sigma T6199) in block overnight at 4℃. Following primary incubation, the cells were washed with 0.1% PBS-T and incubated in 1:2,000 AlexaFluor 488 goat anti-rabbit and 1:2,000 AlexaFluor 568 goat anti-mouse secondary antibodies with 1:1,000 DAPI in block for 1 hour. The cells were then washed with 0.2% PBS-T prior to mounting with ProLong Diamond anti-fade (ThermoFisher). Immunofluorescence microscopy analysis was conducted using a Nikon Ti-E confocal microscope equipped with a spinning disc module. All confocal images were acquired using a ×100 oil-immersion objective, and z-stacks were acquired with a step size of 0.2 µM.

### Image processing and analysis

ImageJ (National Institutes of Health) was used for all confocal image processing. To determine the relative fluorescence intensity of acK40 tubulin following DMSO or AGK2 treatment, confocal images of cell midplanes were acquired using a ×60 objective. Cells were outlined using the Freehand ROI tool and the cell area, integrated density, and average fluorescence intensity for tubulin and acK40 tubulin signals were quantified for all cells. Corrected total cell fluorescence (CTCF) was calculated after background signal subtraction using CTCF = integrated density – (area of selected cell × mean fluorescence of background readings).

### Supernatant and cell-associated virus titer

To determine IUs produced during infection, cell culture supernatant from infected cells was collected at 120 or 144 HPI. To determine cell-associated virus titer, cells were washed with PBS and collected by scraping into complete growth media. The cells were lysed by subjecting them to one freeze-thaw cycle and bath sonicating to release cell-associated virus. To generate the virus titer reporter plate, cells were seeded in 96-well dishes. Virus-containing media collected from either cell culture supernatant or cell-associated samples were used to infect the reporter plate in a dilution series (0.1–0.001). After 24 hours, reporter plate cells were washed with PBS and permeabilized with ice-cold methanol at −20℃ for 15 min. Following permeabilization, the cells were washed with PBS and blocked with 3% BSA in PBS + 0.2% Tween 20 (PBST) for 1 hour. After blocking, the cells were incubated in 1:40 mouse anti-IE1 (clone 1B12, gift from Thomas Shenk) in 0.3% BSA in PBST for 1 hour. Following primary antibody incubation, the cells were washed with PBST and incubated with 1:1,000 goat anti-mouse AlexaFluor 488 and 1:1,000 DAPI in 0.3% BSA in PBST for 1 hour. The cells were then washed with PBST, and the number of infected cells was quantified via immunofluorescence detection of IE1 using the Operetta imaging system (PerkinElmer).

### HCMV genome and particle:IU quantification

To collect cell-associated viral and nuclear genomes, cells were washed with cold PBS, scraped from the plate in complete growth medium, and frozen at −80℃. The cells were lysed using five rounds of 10-second sonication cycles, icing between rounds. To quantify

extracellular virus particles for particle:IU ratio calculation, cell culture supernatant was collected from infected cells at 120 and 144 HPI. A portion of each sample was used to determine supernatant virus titer (IU) by immunofluorescence detection of IE1, as described above. To remove non-packaged viral DNA from extracellular samples, the cell culture supernatant was treated with DNase I for 10 min at 37°C before increasing the temperature to 75°C for 10 min to inactivate the DNase I. Equal volumes of cell lysates were added to DNA resuspension buffer (400 mM NaCl, 10 mM Tris [pH 8], 10 mM EDTA) containing 40-μg/mL proteinase K and 0.2% SDS. Samples were incubated overnight at 37°C to allow for proteinase K-mediated digestion of cellular protein and the viral capsid. DNA was then recovered by phenol:chloroform extraction and ethanol precipitation. Purified DNA was resuspended in RNase-free $H_2O$. qPCR-based genome abundance quantification was performed using SYBR green PCR master mix (Life Technologies) and primers specific to the HCMV IE1 gene (forward primer: 5′-TCGTTGCAATCCTCGGTCA -3′, reverse primer: 5′-ACAGTCAGCTGAGTCTGGGA-3′) and the nuclear β2-microglobulin (B2M) gene (5′-TGCTGTCTCCATGTTTGATGTATCT-3′, reverse primer: 5′-TCTCTGCTCCCCAC CTCTAAGT-3′). Data acquisition was performed using the ViiA7 Real-Time PCR System (Applied Biosystems). pCR2.1 TOPO vectors containing the qPCR-targeted regions of each gene were used to generate standard curves for determining absolute viral and nuclear genome abundances in samples. To quantify viral genomes per cell, intracellular viral genome abundance was normalized to B2M abundance. To determine particle:IU ratios, the extracellular viral genome abundance was divided by the IU abundance for each sample.

## Immunoaffinity purification-mass spectrometry (IP-MS) analysis

### IP and protein digestion

For analysis of SIRT2 interactions during infection, uninfected cells or infected cells at 0, 6, 12, and 24 HPI were collected by washing and scraping into cold PBS. For analysis of the impact of AGK2 on SIRT2 interactions, cells treated with DMSO or 10 μM AGK2 were collected at 48 HPI. The cells were centrifuged to collect cell pellets, which were washed by resuspending in cold PBS followed by another round of centrifugation to recollect cell pellets. The PBS was removed from cells, and 100× HALT protease and phosphatase inhibitor was added (78438, ThermoFisher). The cells were snap frozen in liquid nitrogen and stored at −80°C prior to lysis. Frozen cell pellets were thawed on ice and then suspended in 500 μL of lysis buffer consisting of 20 mM HEPES KOH, pH 7.4, 110 mM potassium acetate, 2 mM $MgCl_2$, 0.1% Tween-20, 1 μM $ZnCl_2$, 1 μM $CaCl_2$, 1% Triton-X, 0.5% deoxycholate, 250 mM NaCl, 1:1,000 universal nuclease, and 1:100 HALT protease and phosphatase inhibitor cocktail. Samples were incubated on ice for 5 min and then vortexed on medium speed in rounds of 15 seconds, icing between rounds, until the cell pellet was completely dissolved. Insoluble debris was pelleted by centrifuging at 10,000 × $g$ for 10 min at 4°C, and the protein concentration of the soluble fraction was determined by BCA. IP samples were generated by aliquoting 500 μg of each sample and bringing to a total volume of 500 μL with lysis buffer (1-mg/mL total protein concentration).

Prior to performing the IP, 4 μg of anti-SIRT2 antibody (Abcam, ab211033) or rabbit IgG was pre-conjugated to 25 μL magnetic protein A/G bead slurry (88802, Thermo-Fisher) in 20 mM HEPES KOH, pH 7.4, 110 mM potassium acetate, 2 mM $MgCl_2$, 0.1% Tween-20, 1 μM $ZnCl_2$, and 1 μM $CaCl_2$ by rotating for 1 hour at 4°C. The beads were collected using a magnetic bar and rinsed with wash buffer (20 mM HEPES KOH, pH 7.4, 110 mM potassium acetate, 2 mM $MgCl_2$, 0.1% Tween-20, 1 μM $ZnCl_2$, 1 μM $CaCl_2$, 1% Triton-X, 0.5% deoxycholate, and 250-mM NaCl) and then resuspended in 25 μL of wash buffer. Twenty-five microliters of conjugated magnetic beads was added to each IP sample, and samples were rotated for 1 hour at 4°C. Beads were collected using a magnetic strip and washed three times with wash buffer and two times with purified water. Proteins were eluted from the beads by heating in 100 μL of TEL buffer (106 mM

Tris HCl, 141 mM Tris base, 2% LDS, 0.5 mM EDTA) for 10 min at 70°C and then shaking for 10 min. Eluates were collected from the beads and reduced and alkylated with 25 mM TCEP (77720, ThermoFisher) and 50 mM chloroacetamide at 70°C for 20 min.

All IP-MS samples were digested and prepared for mass spectrometry analysis using S-Trap (Protifi, C02-micro-80) on-column digestion by following the manufacturer's protocol. Briefly, samples were acidified to a final concentration of 1.2% phosphoric acid, mixed with S-Trap binding buffer (90% methanol, 100-mM triethanolamine bicarbonate [TEAB], pH 7.1) and applied to the S-Trap column. Columns were centrifuged at 4,000 × $g$ for 30 seconds to remove the sample buffer and then washed five times with S-Trap binding buffer. On-column digestion was performed overnight at 37°C using a 1:50 trypsin:protein ratio in 100 µL of 25 mM TEAB (pH 8). Peptides were eluted from the column with sequential addition of 25 mM TEAB (pH 8), 0.2% formic acid (FA), and 50% acetonitrile (ACN) in 0.2% FA. Peptides were concentrated to near dryness via vacuum centrifugation, resuspended in 1% FA/1% ACN, and analyzed by liquid chromatography tandem mass spectrometry (LC-MS/MS).

## Mass spectrometry analysis by data-dependent acquisition

Samples were analyzed via nano-LC-MS/MS using a Dionex Ultimate 3000 nanoRSLC coupled to a Q Exactive HF mass spectrometer (ThermoFisher). Peptides were separated over a 150-min gradient (4% B–50% B) with a 250-nL/min flow rate by reverse-phase chromatography on an EASY-Spray HPLC Column (250-mm length, 2-µm particle size, 75-µm diameter; ThermoFisher). For analysis of the impact of AGK2 treatment on SIRT2 protein interactions, peptides were separated over a 150-min gradient (5% B–30% B) with a 250-nL/min flow rate by reverse-phase chromatography on a 50-cm column packed in-house with 1.9 µM ReproSil-Pur C18-AQ (Dr. Maisch, GmbH). Mobile phase A consisted of 0.1% formic acid in water, and mobile phase B consisted of 0.1% formic acid in 97% ACN. An MS1 scan was performed at a resolution of 120,000 across a mass range of 350–1,800 with an automatic gain control (AGC) of 3e6 and a maximum (max) injection time of 30 ms. Data-dependent MS2 scans of the top 20 ions followed each MS1 scan with collision-induced dissociation fragmentation, resolution of 15,000, an AGC of 1e5, max injection time of 25 ms, a minimum signal of 2,500, isolation width of 1.2, normalized collision energy of 28%, and dynamic exclusion of 45 s.

The MS/MS data were analyzed by Proteome Discoverer 2.4 (ThermoFisher) using a FASTA file containing human and herpesvirus protein sequences and common contaminants. The Spectrum Files RC node and Minora Feature Detector nodes were used to perform offline mass recalibration and label-free MS1 quantitation, respectively. MS/MS spectra were analyzed using Sequest HT for forward and reverse searches to determine FDR. Sequest was run with settings for a fully tryptic search with a maximum of two missed cleavages, precursor mass tolerance of 4 ppm, fragment mass tolerance of 0.02 Da, static carbamidomethylation of cysteine, dynamic deamidation of asparagine, dynamic oxidation of methionine, dynamic pyroglutamic acid formation, dynamic N-terminal acetylation, and dynamic loss of methionine plus acetylation of the protein N-terminus. The matched spectra were validated by the percolator node with 1% FDR. Only unique or razor peptides were used for quantification and RT-dependent normalization, and low abundance-based resampling was performed across samples. Two consensus files were created for the data, one for label-free quantification of proteins enriched in the SIRT2 versus control IgG IP samples and one for bait normalization of enriched proteins using SIRT2 abundance.

## Bioinformatic analysis of protein-protein interaction data

To determine interaction specificity, the $\log_2$ fold-change protein abundance for SIRT2 isolations relative to IgG control isolations was plotted against the $P$ value (calculated using Student's $t$-test with homoskedastic variance and two tails). Proteins were considered to be high-confidence interactors if they passed a $\log_2$ fold-change

abundance threshold of 4 with a *P* value less than 0.05 at any analyzed time point. Protein abundances were normalized to SIRT2 (bait) abundance for each sample, and individual proteins were scaled by predicted tryptic peptide abundance, determined using Perseus. Functional interaction networks were generated as STRING (79) networks in Cytoscape v.3.8.0. Protein localization analysis was performed by manual curation from UniProt Consortium and GO annotation databases. Functional enrichment analysis was performed with Panther, using the statistical overrepresentation test, GO biological process (complete) database, and default settings. Principal component analysis was performed using ClustVis.

## Acetylome and cellular proteome analysis

### Cellular proteome and acetylome sample preparation

To analyze the impact of SIRT2 on cellular proteome abundances and acetylation levels, infected cells were collected at 0, 12, and 24 HPI by washing and scraping into cold PBS. The cells were centrifuged to collect cell pellets, washed once with PBS, and centrifuged again to recollect cells. The PBS was aspirated from cells, and 100× HALT protease and phosphatase inhibitor was added (78438, ThermoFisher). The cells were snap frozen in liquid nitrogen and stored at −80°C prior to preparation for MS analysis. Sample preparation for proteome and acetylome analysis was performed as previously described in reference (11, 128). Briefly, cell pellets were thawed on ice, resuspended in lysis buffer (50-mM Tris-HCl, pH 8, 100-mM NaCl, 0.5-mM EDTA, 4% SDS), and lysed by repeated rounds of boiling and sonication using a cup horn sonicator. Samples were reduced and alkylated with 25-mM TCEP (77720, ThermoFisher), and 50-mM chloroacetamide and proteins were extracted using methanol-chloroform precipitation. Protein pellets were resuspended in 25-mM HEPES, pH 8.2, at a protein concentration of 0.5 mg/mL (determined by BCA assay), and trypsin digestion was performed overnight at 37°C with constant rocking. Two additions of trypsin at a 1:200 trypsin:protein mass ratio were performed 5 hours apart. Following trypsin digestion, samples were acidified to 1% trifluoroacetic acid (TFA) and incubated on ice for 15 min. Insoluble material was removed by centrifugation at 3,700 *g* for 10 min, and desalting was performed using an Oasis Column (Waters) per manufacturer instructions [see also reference (11, 128)]. Samples were frozen in liquid nitrogen and dried using a combination of lyophilization and vacuum centrifugation.

Acetyl lysine immunoaffinity purification was performed using PTMScan Acetyl-Lysine Motif (Ac-K) Kit #13416 from Cell Signaling, following the manufacturer's protocol and modifications detailed in references 11, 128. Fifty micrograms of each sample was reserved for whole proteome analysis prior to acetyl lysine immunoaffinity enrichment. Following the IP, acetylated peptides were eluted from the beads using two additions of 0.15% TFA and incubation at room temperature for 10 min with flicking every 2–3 min. Whole cell proteome samples and acetyl-peptide enrichment samples were acidified to 1% TFA and desalted using SDB-RPS StageTips. Peptides were eluted from the StageTips with 5% ammonium hydroxide/80% ACN, concentrated to near dryness with vacuum centrifugation, and resuspended in 1% FA/1% ACN for LC-MS/MS analysis.

### Mass spectrometry analysis by data-dependent acquisition

Proteome and acetylome samples were analyzed via nano-LC-MS/MS using a Dionex Ultimate 3000 nanoRSLC coupled to a Q Exactive HF mass spectrometer (ThermoFisher). Peptides were separated over a 150-min gradient (3% B–35% B) with a 250-nL/min flow rate by reverse-phase chromatography on an EASY-Spray HPLC Column (500-mm length, 2-µm particle size, 75-µm diameter; ThermoFisher). Mobile phase A consisted of 0.1% formic acid in water, and mobile phase B consisted of 0.1% formic acid in 97% ACN. An MS1 scan was performed at a resolution of 120,000 across a mass range of 350–1,800 with an AGC of 3e6 and max injection time of 30 ms, For acetylome samples, data-dependent MS2 scans of the top 10 ions followed each MS1 scan with collision-induced

dissociation fragmentation, resolution of 30,000, an AGC of 1e5, max injection time of 150 ms, a minimum signal of 4,200, isolation width of 1.6, normalized collision energy of 28%, and dynamic exclusion of 30 seconds. For proteome samples, data-dependent MS2 scans of the top 10 ions followed each MS1 scan with collision-induced dissociation fragmentation, resolution of 15,000, an AGC of 1e5, max injection time of 25 ms, a minimum signal of 2,500, isolation width of 1.2, normalized collision energy of 28%, and dynamic exclusion of 30 seconds.

The MS/MS data were analyzed by Proteome Discoverer 2.4 (ThermoFisher) using a FASTA file containing human and herpesvirus protein sequences and common contaminants. The Minora Feature Detector and Spectrum Files RC nodes were used to perform label-free MS1 quantitation and mass recalibration. Sequest was run with a full tryptic search and two maximum missed cleavages, precursor mass tolerance of 4 ppm and fragment mass tolerance of 0.02 Da. Included modifications were static cysteine carbamidomethylation, dynamic lysine acetylation, dynamic asparagine deamidation, dynamic methionine oxidation, dynamic N-terminal acetylation, and dynamic loss of methionine plus acetylation of the protein N-terminus. The matched spectra were validated with percolator (1% FDR). Only unique or razor peptides were used for quantification. For whole proteome samples, normalization was performed in PD2 by total peptide amount. Acetyl-IP samples were not normalized using PD2.

### Bioinformatic analysis

Whole proteome data were normalized in PD2, and principal component analysis was performed using ClustVis. Log$_2$ fold change abundances were clustered using k-means, and a heatmap was generated using the Seaborn Python package. The optimal number of clusters was determined using the elbow method. K-means clustering was performed with Cluster3.0 with organization by gene, k = 7 clusters, and 100 iterations. Results were exported using Java Treeview. Enriched gene ontology terms for each cluster were determined using the Functional Annotation Chart tool of DAVID with default parameters and human proteomic data as background. Enriched terms were selected from GO biological processes.

Acetylome data were median normalized in Excel and missing values were imputed using Perseus. To identify peptides with AGK2-regulated acetylations, the log$_2$ fold change peptide abundance for AGK2 treatment relative to DMSO treatment was calculated and plotted against the $P$ value (calculated using Student's $t$-test with homoskedastic variance and two tails). Acetylated peptides were further investigated if they (i) demonstrated a statistically significant (i.e., $P$ value of <0.05) increase in acetylation with AGK2 treatment or (ii) were detected for both AGK2-treated samples but not for either DMSO-treated sample at a given time point. Principal component analysis was performed using ClustVis, and protein localization was determined by manual curation from UniProt Consortium and GO annotation databases. For motif analysis, acetyl-peptide sequences were extended with PeptidExtender, using the *Homo sapiens* proteome as background to obtain peptides with six amino acids flanking each side of the acetylated lysine. Motifs were generated using IceLogo with a $P$ value of 0.01 and the *Homo sapiens* proteome database as a reference set (129). Gene ontology enrichment analysis was performed with Panther using the statistical overrepresentation test, GO biological process (complete) database, and default settings.

### Flow cytometry-based cell cycle profiling

One hour prior to sample collection, 10 µM EdU was added to the cell culture media to label newly synthesized DNA. Following EdU labeling, cells were washed with PBS and collected by trypsinization followed by centrifugation. Cell pellets were washed with PBS and fixed by incubating in 4% PFA for 20 min. Fixed cells were washed with PBS, collected by centrifugation, and permeabilized with 0.25% Triton X-100 in PBS for 15 min on ice. Fixed and permeabilized cells were washed with wash solution (1% FBS in PBS) and collected by centrifugation. To label S-phase cells, EdU that was incorporated into newly

synthesized DNA was conjugated with azide-AlexaFluor-647 (ClickChemistryTools) by incubating samples in Click-IT staining solution (10 mM sodium ascorbate, 2 mM CuSO$_4$, 10 µM Az-647 in PBS) for 1 hour at room temperature, mixing every 15 min. Following the 1-hour incubation, cells were washed with wash solution and collected by centrifugation. To stain for phosphorylated histone H3 Ser10 (a mitotic cell marker), a pH3-Ser10 antibody pre-conjugated to biotin (16189MI, ThermoFisher) was used. Samples were incubated for 1 hour in a 1:100 dilution of pH3-Ser10 antibody in wash solution with mixing every 15 min. Following primary antibody incubation, samples were washed with wash solution, collected by centrifugation, and stained with 1:100 Streptavidin-PE-Cy7 (SA1012, ThermoFisher) secondary antibody and 1:1,000 DAPI in wash solution for 1 hour while mixing every 15 min. Fully stained cells were washed with wash solution, passed through a cell strainer, and analyzed by flow cytometry. Flow cytometry analysis was performed using a BD LSRII Multi-Laser Analyzer with HTS and FACSDIVA software. Data analysis was performed using FCSExpress and FloJo software (BD Biosiences).

## CDK2-IN-4 treatment and infection conditions

To assess whether CDK2 activity contributes to the decrease in HCMV titer observed with AGK2 treatment, cells were treated 12 hours prior to infection with either DMSO (vehicle control) or 5 µM CDK2-IN-4 (MedChemExpress [115[) in complete growth media. After the 1-hour incubation with HCMV inoculated media (not containing pharmacological inhibitors), complete growth media containing DMSO or 2.5-µM AGK2 were added to the cells, and supernatant virus was collected at 144 HPI. A final growth media DMSO concentration of 0.1% was used for all treatments.

## Generation of CDK2 acetyl mimic stable cell lines

Sequences encoding WT CDK2-3×FLAG, acetyl mimic CDK2(K6Q)-3×FLAG, or lysine charge mimic CDK2(K6R)-3×FLAG were cloned into the pLXSN Retroviral Vector (Takara). To produce retrovirus vectors, Phoenix-Ampho cells (ATCC, CRL-3213) were plated in 10-cm dishes (one dish per CDK2-3×FLAG construct) and transfected at 70% confluency with 30 µg pLXSN Vector and 54 µL Xtremegene in OptiMEM, replacing the media 6 hours post-transfection (hpt) with complete growth media. Cell culture supernatant containing retrovirus was collected at 48 and 72 hpt. Retrovirus media were centrifuged, passed through a 0.45-µm filter, and used to reverse-transduce a 15-cm plate of MRC5 cells. Transduced MRC5 cells were selected by splitting two times into complete growth media containing G418 (400 µg/mL) prior to performing experiments.

## Parallel reaction monitoring (PRM)-based quantification of SIRT2 and HCMV protein abundances

### Sample preparation for MS analysis

To quantify SIRT2 and viral protein abundances throughout HCMV infection, infected cells were harvested at 12, 24, 48, 72, and 96 HPI by washing and scraping into cold PBS followed by centrifugation to collect cell pellets. Cells were washed twice with cold PBS, spiked with 100× HALT protease and phosphatase inhibitor ( 78438, ThermoFisher, 78438), snap frozen in liquid nitrogen, and stored at −80°C. Frozen cell pellets were briefly thawed on ice and then lysed in 5% SDS, 25-mM TCEP (77720, ThermoFisher), and 50mM chloroacetamide using repeated rounds of boiling followed by cup horn sonication. Proteins were recovered via methanol-chloroform precipitation (as described in reference 130), and protein concentration was determined by BCA assay. Proteins were resuspended in 100 mM HEPES (pH 8.2) at a 0.5-µg/µL concentration and digested for 16 hours at 37°C with MS-grade Pierce trypsin (90057, ThermoFisher) at a 1:50 trypsin:protein mass ratio. Digested samples were acidified to 1% TFA, desalted using 3M Empore C18 Extraction Disks (14-386-2, ThermoFisher), and eluted from the disks with 70% ACN/ 0.5% FA. Peptides were concentrated to near dryness via vacuum centrifugation and resuspended in 1% FA/1% ACN for LC-MS/MS analysis.

### Targeted MS analysis by PRM

Targeted MS analysis was performed via LC-MS/MS using a Dionex Ultimate 3000 nanoRSLC coupled to a Q Exactive HF mass spectrometer (ThermoFisher). Peptides were separated by reverse-phase chromatography on an EASY-Spray HPLC Column (250-mm length, 2-µm particle size, 75-µm diameter; ThermoFisher) using a 60-min gradient (2%–38% B) with 250-nL/min flow rate. Mobile phase A consisted of 0.1% formic acid in water, and mobile phase B consisted of 0.1% formic acid in 97% ACN. The PRM method consisted of targeted MS2 scans recorded in profile mode performed at a resolution of 30,000, with an AGC target of 1e5, maximum injection time of 200 ms, isolation window of 1.2, and normalized collision energy of 27% controlled by a peptide inclusion list derived from our TRUSTED targeted MS assay (120). An MS1 scan was performed at a resolution of 15,000 across a mass range of 400–2,000 with an AGC of 3e6, and max injection time of 15 ms.

### Quantification of protein abundances

Label-free quantitation was performed using Skyline software for targeted proteomics (131), using the method detailed in reference 120. Briefly, a summed area under the curve of three transitions per peptide was used for quantitation. Normalization across samples and replicates was performed using MS1 intensity-based normalization. The MS1 intensity for each file was determined using RawMeat (Vast Scientific) and used to normalize peptide peak areas relative to other samples belonging to the same replicate. Following MS1-based normalization, individual peptides were normalized to their average abundance across all time points/sample conditions. After performing MS1 and mean normalizations, if there were multiple peptides per protein, they were averaged. Average protein abundance was then calculated across replicates, and proteins were organized by temporality as in reference 120. To perform Euclidean distance analysis, the distance for a given HCMV protein was determined as $d(p,q) = \sqrt{(q_i - p_j)^2}$, where $q$ = AGK2 samples, $p$ = DMSO samples, and $i$ and $j$ are time points of infection. The average Euclidean distance was calculated for each time point comparison, and the values were min-max scaled from 0 to 1. Principal component analysis was performed using ClustVis.

## Multi-step growth curve analysis

To perform the multi-step HCMV growth curve analysis following AGK2 treatment, cells were plated in three replicate 96-well plates per treatment condition and treated at 70% confluency with DMSO or 2.5 µM AGK2. Cells were infected with serial dilutions of HCMV ($10^{-2}$ to $10^{-9}$) for 1 hour, after which viral inoculum was replaced with DMSO- or AGK2-treated cell culture media. Plate wells were observed visually for viral plaque formation at 5-, 10-, and 16 DPI. TCID50/mL was calculated by the Spearman and Karber algorithm as described in reference 132.

## Quantification and statistical analysis

Excel or GraphPad Prism v.8 software was used for statistical analysis. The numbers of replicates and statistical tests used for each data set are indicated in the corresponding figure legend. Error bars show standard deviation and bars show arithmetic mean, unless otherwise indicated. Statistical significance is indicated with asterisks in figures: * for $P \leq$ 0.05, ** for $P \leq$ 0.01, *** for $P \leq$ 0.001, and **** for $P \leq$ 0.0001.

## ACKNOWLEDGMENTS

We thank Gary Laevsky (Princeton University Molecular Biology Confocal Imaging Facility) and Christina DeCoste (Princeton University Molecular Biology Flow Cytometry Resource Facility) for instrument use and technical support. The Flow Cytometry Resource Facility is partially supported by the Rutgers Cancer Institute of New Jersey NCI-CCSG P30CA072720-5921, and the FACSymphony Flow Cytometery is funded by

the S10 Shared Instrumentation Grant S10OD028592. We thank Todd Greco and Josiah Hutton III for MS support and Thomas Shenk for HCMV strains and antibodies.

We are grateful for funding from the National Institutes of Health (NIH) (GM114141 and AI174515) and Stand Up to Cancer Convergence 3.1416 to I.M.C., an NIH Ruth L. Kirschstein NRSA fellowship (F31AI154796) to C.N.B., New Jersey Commission on Cancer Research Pre-Doctoral Fellowship (COCR23PRF019) to M.D.T., an NSF Graduate Research Fellowship (DGE1148900) to H.G.B., and a National Institute of General Medical Sciences training grant (T32GM007388) to C.N.B., M.D.T., and H.G.B. The funders had no role in study design, data collection and interpretation, or the decision to submit the work for publication.

C.N.B. and I.M.C. designed the research. C.N.B., J.L.J., M.D.T., J.E.E., H.G.B., and Y.F.A. performed the experiments and analyzed the data. C.N.B. and I.M.C. wrote the manuscript.

## AUTHOR AFFILIATION

[1]Department of Molecular Biology, Princeton University, Lewis Thomas Laboratory, Washington Road, Princeton, New Jersey, USA

## AUTHOR ORCIDs

Joshua L. Justice http://orcid.org/0000-0003-1884-8555
Matthew D. Tyl http://orcid.org/0000-0002-6543-9028
Ileana M. Cristea http://orcid.org/0000-0002-6533-2458

## FUNDING

| Funder | Grant(s) | Author(s) |
|---|---|---|
| HHS | NIH | National Institute of General Medical Sciences (NIGMS) | GM114141 | Ileana M. Cristea |
| HHS | NIH | National Institute of Allergy and Infectious Diseases (NIAID) | AI174515 | Ileana M. Cristea |
| HHS | NIH | National Institute of General Medical Sciences (NIGMS) | T32GM007388 | Cora N. Betsinger |
| | | Matthew D. Tyl |
| | | Hanna G. Budayeva |
| National Science Foundation (NSF) | DGE1148900 | Hanna G. Budayeva |
| HHS | NIH | NIAID | Division of Intramural Research, National Institute of Allergy and Infectious Diseases (DIR, NIAID) | F31AI154796 | Cora N. Betsinger |
| New Jersey Commision on Cancer Research | COCR23PRF019 | Matthew D. Tyl |
| EIF | Stand Up To Cancer (SU2C) | Convergence 3.1416 | Ileana M. Cristea |

## AUTHOR CONTRIBUTIONS

Cora N. Betsinger, Conceptualization, Formal analysis, Investigation, Writing – original draft | Joshua L. Justice, Formal analysis, Visualization | Matthew D. Tyl, Formal analysis, Funding acquisition | Julia E. Edgar, Formal analysis, Investigation | Hanna G. Budayeva, Formal analysis, Funding acquisition, Investigation | Yaa F. Abu, Investigation | Ileana M. Cristea, Conceptualization, Funding acquisition, Project administration, Supervision, Writing – original draft

## DATA AVAILABILITY

Mass spectrometry proteome raw data and result files are deposited in the ProteomeX-change Consortium via the PRIDE (131) partner repository with identifier PXD041171. Skyline data files containing results from targeted MS analysis of HCMV protein

abundances are deposited in PanoramaWeb and will be made publicly available at the time of publication. Processed IP-MS and acetylome data sets are also included as supplementary tables (Tables S1 and S2).

## ADDITIONAL FILES

The following material is available online.

### Supplemental Material

**Fig. S1 (mSystems00510-23-s0001.tif).** Validation of AGK2 efficacy in human fibroblast cell culture model.

**Fig. S2 (mSystems00510-23-s0002.tif).** Assessment of SIRT2 immunoaffinity purification efficiency and IP-MS data quality.

**Fig. S3 (mSystems00510-23-s0003.tif).** Assessment of SIRT2 immunoaffinity purification efficiency and IP-MS data quality.

**Fig. S4 (mSystems00510-23-s0004.tif).** Analysis of proteins with acetylation sites that increased in abundance following SIRT2 inhibition.

**Fig. S5 (mSystems00510-23-s0005.tif).** Analysis of cellular proteome alterations driven by SIRT2 inhibition.

**Fig. S6 (mSystems00510-23-s0006.tif).** Assessment of the impact of SIRT2-mediated cell cycle regulation on HCMV replication.

**Table S1 (mSystems00510-23-s0007.xlsx).** List of cellular and viral proteins detected in the endogenous SIRT2 Immunoaffinity Purification-Mass Spectrometry Analyses.

**Table S2 (mSystems00510-23-s0008.xlsx).** Acetylated peptides and proteins detected in our acetylome and whole-cell proteome analysis.

### Open Peer Review

**PEER REVIEW HISTORY (review-history.pdf).** An accounting of the reviewer comments and feedback.

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
