## [Reviewer comments · mSystems]

Sirtuin 2 promotes human cytomegalovirus replication by regulating cell cycle progression

Cora Betsinger, Joshua Justice, Matthew Tyl, Julia Edgar, Hanna Budayeva, Yaa Abu, and Ileana Cristea

Corresponding Author(s): Ileana Cristea, Princeton University

Review Timeline:

Submission Date:	May 19, 2023
Editorial Decision:	June 23, 2023
Revision Received:	August 10, 2023
Accepted:	September 28, 2023

Editor: Joshua Elias

Reviewer(s): The reviewers have opted to remain anonymous.

Transaction Report:

DOI: <https://doi.org/10.1128/msystems.00510-23>

June 23, 2023

Prof. Ileana M Cristea
Princeton University
Dept. of Molecular Biology
210 Lewis Thomas Laboratory
Washington Road
Princeton, NJ 08544-1014

Re: mSystems00510-23 (Sirtuin 2 promotes human cytomegalovirus replication by regulating cell cycle progression)

Dear Prof. Ileana M Cristea:

Thank you for submitting your manuscript to mSystems. We have completed our review and I am pleased to inform you that, in principle, we expect to accept it for publication in mSystems. However, acceptance will not be final until you have adequately addressed the reviewer comments.

Preparing Revision Guidelines

Please return the manuscript within 60 days; if you cannot complete the modification within this time period, please contact me. If you do not wish to modify the manuscript and prefer to submit it to another journal, please notify me of your decision immediately so that the manuscript may be formally withdrawn from consideration by mSystems.

Sincerely,

Joshua Elias

Editor, mSystems

Journals Department
Reviewer comments:

Reviewer #1 (Comments for the Author):

Betsinger et al. study the role of SIRT2 and specifically its deacetylase activity in the progression of human cytomegalovirus infection in cell lines.

In light of SIRT2's multifunctional nature, the authors mainly use the SIRT2-specific deacetylase inhibitor AGK2 (as opposed to genetic targeting) as the main tool to characterize that function in particular.

Using a multitude of methods, including FACS analysis, confocal microscopy, protein-protein interaction mapping, protein acetylation profiling, and whole proteome measurements, the authors ultimately conclude that SIRT2's deacetylation activity is required in the early stage of infection because inhibition of that activity drives cells into S phase, which is not conducive to HCMV replication.

Overall, the paper is well-written and will be of interest to the virology community.

Below, I have made a series of comments that I hope will improve the presentation of the authors' findings, making them easier to understand for most readers.

Major:

1. The exact experimental distinction between "mock" and "0h" samples is unclear throughout the paper. Does "0h" refer to the time point exactly at the end of the 1h incubation with inoculum? If so, it might be relabeled as "1h"?

"Mock" and "0h" may be interpreted as one and the same condition, experimentally speaking, so it's confusing that they show substantial differences in some cases (e.g. Fig. S2E, or the statement in line 353 that "acetylation increased 3.6-fold at 0 HPI"). In all experiments that involve the calculation of ratios, it needs to be explicitly specified what the reference condition is (mock or 0h).

2. The authors should make all mass spec raw data available via one of the commonly used data repositories.

Minor:

1. What is your source of subcellular localization annotation? I didn't find it referenced anywhere.

2. Fig 4 E/F:

Instead of panel F, you could add a third panel to E with the respective ratios of each of the proteins (similar to 3E). This would assign a protein label to what are currently unlabeled lines in panel F and adds visual continuity.

3. Fig S1, panel E:

It's unclear what the "control +" condition represents

4. Fig 2D, 3G

The pie charts are oddly non-symmetrical; the "2 o'clock" divider doesn't point at the origin. If there's a straightforward fix, please fix this!

(Moreover, I must say that I find the pie chart visualization not ideal, as it implies circularity (coming back to mock after 24h). Small bar graphs or 5 color-coded squares could be more appropriate.)

5. Nobre et al. (PMID 31873071) found SIRT2 interacting with UL133. This should be referenced among known associations of viral proteins with SIRT2.

Reviewer #2 (Comments for the Author):

The manuscript "Sirtuin 2 promotes human cytomegalovirus replication by regulating cell cycle progression" by Betsinger et al. describes how human SIRT2's deacetylase activity serves a pro-viral function for HCMV by regulating cell cycle. To uncover the

mechanisms underlying this pro-viral effect, the authors utilized mass spectrometry to define the protein interactions and deacetylase substrates of SIRT2 at the early stage of HCMV infection. Using multi-disciplinary approaches, the authors demonstrated that SIRT2 interacts and modulates CDK2, and that SIRT2 deacetylase activity governs the G1-S phase transition, which facilitates HCMV replication.

This study addresses the debated functions of SIRT2s during viral infection and expands the understanding of the molecular mechanisms underlying host-virus interactions. Overall, the multi-pronged strategy is well described, and the conclusions are well supported by high-quality mass spec data. The complex but clear graphics allows for effortless understanding of the content. Nonetheless, there are a few minor issues that would further improve the quality of this work if addressed properly.

Suggestions:

1. AGK2 is a quinoline compound that targets the nicotinamide-binding site of SIRT2. It's unclear from this manuscript whether the other activities of SIRT2 are also impacted by AGK2. Admittedly, this is experimentally challenging, so it would be helpful to briefly discuss the potential side effect of AGK2 treatment in the discussion section. Alternatively, relevant references could be included.
2. The authors have systematically characterized SIRT2 interactions and how AGK2 alters cellular acetylome. However, it remains elusive whether AGK2 treatment could impact the interactions. This piece of evidence, when overlapped with the SIRT2 temporal interaction network and acetylome data, may explain the dynamic acetylation levels of these proteins and consolidate the regulation of SIRT2 on cell cycle. This could be done by either MS or WB.
3. Does AGK2 treatment impact SIRT2 protein abundance during HCMV infection?
4. Even though AGK2 did not induce cell death, it can still disrupt cell replication. Therefore, the authors should count the cell numbers at different stages during HCMV infection with or without AGK2.

Minor edits:

1. Fig. S1B. The actin bands are slightly overexposed, which might affect the interpretation of the AcK40 tubulin levels.
2. Page 14 line 334. "28 of the putative" should be written as "Twenty-eight of the putative".

We thank the reviewers for their constructive feedback on our initial manuscript submission. We have made all the changes requested by the reviewers, performing several new experiments and editing the results and methods sections for clarity. Specifically, we have analyzed the impact of AGK2 treatment on SIRT2 protein abundance and cellular proliferation throughout HCMV infection (new figures S1F-G, S6A). To further confirm CDK2 as a substrate of SIRT2 and investigate the specific contribution of its acetylation, we generated cell lines stably expressing CDK2 K6 acetylation mutants (WT, K6Q, K6R) (new Fig. 4F and S6D). Using these cell lines, we demonstrate that CDK2 K6Q (acetylation mimic) recapitulates our findings generated using AGK2 treatment, showing an increase in S-phase cell population. To clarify the difference between our Mock and 0 HPI time points, we have generated a schematic (Figure 1A) and explained the distinction in the results, methods, and figure legend text. All other comments and suggestions have been addressed with other minor experiments and text additions. We are grateful for the reviewers' remarks, and we believe that by responding to these comments and performing these additional experiments, we have strengthened the impact and improved the presentation of our findings. Below we address the reviewers' comments in depth.

Reviewer #1 (Comments for the Author):

Betsinger et al. study the role of SIRT2 and specifically its deacetylase activity in the progression of human cytomegalovirus infection in cell lines. In light of SIRT2's multifunctional nature, the authors mainly use the SIRT2-specific deacetylase inhibitor AGK2 (as opposed to genetic targeting) as the main tool to characterize that function in particular. Using a multitude of methods, including FACS analysis, confocal microscopy, protein-protein interaction mapping, protein acetylation profiling, and whole proteome measurements, the authors ultimately conclude that SIRT2's deacetylation activity is required in the early stage of infection because inhibition of that activity drives cells into S phase, which is not conducive to HCMV replication. Overall, the paper is well-written and will be of interest to the virology community. Below, I have made a series of comments that I hope will improve the presentation of the authors' findings, making them easier to understand for most readers.

Major:

1. The exact experimental distinction between "mock" and "0h" samples is unclear throughout the paper. Does "0h" refer to the time point exactly at the end of the 1h incubation with inoculum? If so, it might be relabeled as "1h"? "Mock" and "0h" may be interpreted as one and the same condition, experimentally speaking, so it's confusing that they show substantial differences in some cases (e.g. Fig. S2E, or the statement in line 353 that "acetylation increased 3.6-fold at 0 HPI"). In all experiments that involve the calculation of ratios, it needs to be explicitly specified what the reference condition is (mock or 0h).
We apologize for the confusion. Following the convention in the HCMV field, 0 HPI refers to the time point immediately at the end of the period allowed for the attachment of the virus to host cells and the initial entry (i.e., in this case 1 hour). To clarify this workflow, we have included a diagram of our treatment and infection procedure as Fig 1A, and refer to literature in the field that supports this nomenclature convention. We have additionally extended our explanation of our treatment and infection protocol in the results and materials and methods sections, as well as in the legend for Figure 1. For any ratio calculations, we have also now included what the reference condition is in our figure legends.
2. The authors should make all mass spec raw data available via one of the commonly used data repositories.

We fully agree and have already deposited our raw data with our initial submission. Our IP-MS, proteome, and acetylome datasets (raw data and result files) are deposited in ProteomeXchange (identifier PXD041171; Username: reviewer_pxd041171@ebi.ac.uk, Password: qq0YLoOJ). Skyline files for targeted MS datasets are deposited in PanoramaWeb and will be made available at the time of publication.

Minor:

1. What is your source of subcellular localization annotation? I didn't find it referenced anywhere.
Protein localization was determined by manual curation from UniProt and GO annotation databases. This information has now been included in the legends for Figures 3 and S2.
2. Fig 4 E/F: Instead of panel F, you could add a third panel to E with the respective ratios of each of the proteins (similar to 3E). This would assign a protein label to what are currently unlabeled lines in panel F and adds visual continuity.
We agree that this is a better representation of the fold-change data. We have removed the line plots and replaced them with a third heat map with the fold-change data included. This panel is now part of Figure 5A, as this allowed us to include new functional follow-up data on CDK2 in Figure 4 (please see below in additional experiments section).
3. Fig S1, panel E: It's unclear what the "control +" condition represents
We apologize for not including this information in the initial submission. The control (+) condition is a positive control generated during the TUNEL assay by treating cells with DNase I (to generate DNA breaks that are then labeled with the TUNEL assay label solution). We have included descriptions of the positive and negative controls in the Figure S1 legend.
4. Fig 2D, 3G: The pie charts are oddly non-symmetrical; the "2 o'clock" divider doesn't point at the origin. If there's a straightforward fix, please fix this! (Moreover, I must say that I find the pie chart visualization not ideal, as it implies circularity (coming back to mock after 24h). Small bar graphs or 5 color-coded squares could be more appropriate.)
To address this comment, we have now changed the display in Figure 2D and 3G from the original pie charts to donut shapes. While fitting the temporal nature of our experiment, we agree that circular graphs imply circularity; however, we would like to have continuity between the IP-MS data shown in Figure 2D and Figure 3G. To differentiate the acetylome dataset from the interactome dataset in Figure 3G, we believe having charts of different shapes is helpful.
5. Nobre et al. (PMID 31873071) found SIRT2 interacting with UL133. This should be referenced among known associations of viral proteins with SIRT2.
Thank you for bringing this previously known SIRT2 interaction to our attention. This interaction was not found in our dataset, likely due to the specific temporality of this association. Our IP-MS experiment focused on early time points of infection, with our latest time point being 24 HPI, while the SIRT2-UL133 interaction was previously reported at 60 HPI. We have added a discussion of this previously identified SIRT2-pUL133 interaction to our results section, as well as a citation to this manuscript.

Additional New Experiments Included:

We also performed additional functional follow-up experiments to characterize the specific impact of CDK2 activity and acetylation on virus titer and cell cycle progression. Cell cycle

progression is regulated by the inhibition and activation of cyclin-dependent protein kinases (illustrated in new Figure 4D). Among these, CDK2 is the major regulator of cell cycle progression from G1 to S phase. Based on our cell cycle profiling results, our identification of a SIRT2-CDK2 interaction at early time points of infection, and increased CDK2 K6 acetylation following SIRT2 inhibition, we next investigated whether SIRT2 functions through the regulation of CDK2 activity during HCMV infection. To address this, we performed two new experiments:

- a. We inhibited CDK2 activity prior to HCMV infection using the pharmacological inhibitor CDK2-IN-4. Cells either pretreated or not treated with CDK2-IN-4 were infected with HCMV and then subjected to AGK2 treatment at 0 HPI (Fig 4E, S6C). Compared to those not exposed to CDK2 inhibition, pretreatment with CDK2-IN-4 resulted in a 50% increase in virus titer in AGK2-treated cells (Fig 4E). These results indicate that SIRT2-mediated regulation of CDK2 activity is at least in part responsible for the effect on virus titer.
- b. We next sought to confirm the specific involvement of CDK2 K6 acetylation status in our observed cell cycle phenotype upon SIRT2 inhibition. MRC5 fibroblasts stably expressing either wildtype (WT) CDK2-3xFLAG, an acetyl mimic CDK2(K6Q)-3xFLAG, or a lysine charge mimic CDK2(K6R)-3xFLAG construct were generated (Fig S6D). Flow cytometry-based cell cycle profiling of these cell lines recapitulated our findings with SIRT2 inhibition (Fig 4F). Expression of the acetyl mimic CDK2(K6Q) construct increased the proportion of cells in S phase and decreased the proportion of cells in G1 relative to expression of WT CDK2. Expression of the lysine charge mimic CDK2(K6R) construct had no significant impact on the proportion of cells in G1, S, or G2 phase relative to WT CDK2 expression.

Altogether, these findings demonstrate that SIRT2 inhibition via AGK2 treatment results in increased acetylation of CDK2 K6, which induces the cell cycle G1-to-S phase transition.

Reviewer #2 (Comments for the Author):

The manuscript "Sirtuin 2 promotes human cytomegalovirus replication by regulating cell cycle progression" by Betsinger et al. describes how human SIRT2's deacetylase activity serves a pro-viral function for HCMV by regulating cell cycle. To uncover the mechanisms underlying this pro-viral effect, the authors utilized mass spectrometry to define the protein interactions and deacetylase substrates of SIRT2 at the early stage of HCMV infection. Using multi-disciplinary approaches, the authors demonstrated that SIRT2 interacts and modulates CDK2, and that SIRT2 deacetylase activity governs the G1-S phase transition, which facilitates HCMV replication.

This study addresses the debated functions of SIRT2s during viral infection and expands the understanding of the molecular mechanisms underlying host-virus interactions. Overall, the multi-pronged strategy is well described, and the conclusions are well supported by high-quality mass spec data. The complex but clear graphics allows for effortless understanding of the content. Nonetheless, there are a few minor issues that would further improve the quality of this work if addressed properly.

Suggestions:

1. AGK2 is a quinoline compound that targets the nicotinamide-binding site of SIRT2. It's unclear from this manuscript whether the other activities of SIRT2 are also impacted by AGK2. Admittedly, this is experimentally challenging, so it would be helpful to briefly discuss the potential side effect of AGK2 treatment in the discussion section. Alternatively, relevant references could be included.

This is an important point that we have carefully considered at the beginning of our study. To address the reviewer's comment, we have now expanded upon the justification for using AGK2 in the results section of the manuscript. We were particularly interested in the deacetylase activity of SIRT2 based on our previous acetylome analysis during HCMV infection (Murray, Sheng, Cristea; 2018). Therefore, AGK2 was advantageous to us as it has been previously demonstrated to be specific for SIRT2 deacetylase activity, showing no inhibitory activity for SIRT2 defatty-acylase functions (Kawaguchi et al, 2019). AGK2 has also been demonstrated to have high selectivity for SIRT2 (Outeiro et al, 2007), only resulting in slight inhibition of other SIRT proteins at 10-fold higher concentrations than those used in our analyses.

2. The authors have systematically characterized SIRT2 interactions and how AGK2 alters cellular acetylome. However, it remains elusive whether AGK2 treatment could impact the interactions. This piece of evidence, when overlapped with the SIRT2 temporal interaction network and acetylome data, may explain the dynamic acetylation levels of these proteins and consolidate the regulation of SIRT2 on cell cycle. This could be done by either MS or WB.

To address the reviewer's comment that pertains to strengthening our predicted SIRT2 substrates and the effect of SIRT2 on cell cycle progression, we performed two experiments:

1. We investigated SIRT2 protein interactions with and without AGK2 treatment (Figure S6B). We found that AGK2 treatment induced a decrease in the levels of a number of SIRT2 interactions, including CDK2. This observation supports our finding of increased CDK2 acetylation upon AGK2 treatment.
2. We also performed additional functional follow-up experiments to characterize the specific impact of CDK2 activity and acetylation on virus titer and cell cycle progression. Cell cycle progression is regulated by the inhibition and activation of cyclin-dependent protein kinases (illustrated in new Figure 4D). Among these, CDK2 is the major regulator of cell cycle progression from G1 to S phase. Based on our cell cycle profiling results, our identification of a SIRT2-CDK2 interaction at early time points of infection, and increased CDK2 K6 acetylation following SIRT2 inhibition, we next investigated whether SIRT2 functions through the regulation of CDK2 activity during HCMV infection. To address this, we performed two new experiments:
 - a. We inhibited CDK2 activity prior to HCMV infection using the pharmacological inhibitor CDK2-IN-4. Cells either pretreated or not treated with CDK2-IN-4 were infected with HCMV and then subjected to AGK2 treatment at 0 HPI (Fig S6C). Compared to those not exposed to CDK2 inhibition, pretreatment with CDK2-IN-4 resulted in a 50% increase in virus titer in AGK2-treated cells (Fig 4E). These results indicate that SIRT2-mediated regulation of CDK2 activity is at least in part responsible for the effect on virus titer.
 - b. We next sought to confirm the specific involvement of CDK2 K6 acetylation status in our observed cell cycle phenotype upon SIRT2 inhibition. MRC5 fibroblasts stably expressing either wildtype (WT) CDK2-3xFLAG, an acetyl mimic CDK2(K6Q)-3xFLAG, or a lysine charge mimic CDK2(K6R)-3xFLAG construct were generated (Fig S6D). Flow cytometry-based cell cycle profiling of these cell lines recapitulated our findings with SIRT2 inhibition (Fig 4F). Expression of the acetyl mimic CDK2(K6Q) construct increased the proportion of cells in S phase and decreased the proportion of cells in G1 relative to expression of WT CDK2. Expression of the lysine charge mimic CDK2(K6R) construct had no significant impact on the proportion of cells in G1, S, or G2 phase relative to WT CDK2 expression.

Altogether, these findings demonstrate that SIRT2 inhibition via AGK2 treatment results in increased acetylation of CDK2 K6, which induces the cell cycle G1-to-S phase transition.

3. Does AGK2 treatment impact SIRT2 protein abundance during HCMV infection?

To address the reviewer's comment, we have now included western blot and targeted MS analysis of SIRT2 abundance throughout HCMV infection following AGK2 or DMSO treatment (Fig S1F and G). We find that AGK2 treatment results in a slight increase in SIRT2 abundance in uninfected conditions, no significant change in SIRT2 abundance at early and middle stages of infection, and a slight but statistically significant increase in SIRT2 abundance at 96 HPI. Therefore, the acetylome datasets are at timepoints when SIRT2 abundance is not impacted by AGK2 treatment.

4. Even though AGK2 did not induce cell death, it can still disrupt cell replication. Therefore, the authors should count the cell numbers at different stages during HCMV infection with or without AGK2.

To address the reviewer comments, we now include a quantification of cell count at 24 and 144 HPI following AGK2 or DMSO treatment. We do indeed see an increase in cell count following AGK2 treatment at 144 HPI, supporting our finding that AGK2 drives cell cycle progression (Fig S6A).

Minor edits:

1. Fig. S1B. The actin bands are slightly overexposed, which might affect the interpretation of the AcK40 tubulin levels.

We agree that the actin bands are overexposed in this blot. Unfortunately, we were unable to receive the anti-ack40 tubulin antibody to redo this result in time for resubmission; so, we have now included a second western blot for acetylated histone H3K9 (an additional known SIRT2 substrate) in Figure S1.

2. Page 14 line 334. "28 of the putative" should be written as "Twenty-eight of the putative".

Thank you for bringing this to our attention, we have made this edit in the text.

September 28, 2023

Prof. Ileana M Cristea
Princeton University
Dept. of Molecular Biology
210 Lewis Thomas Laboratory
Washington Road
Princeton, NJ 08544-1014

Re: mSystems00510-23R1 (Sirtuin 2 promotes human cytomegalovirus replication by regulating cell cycle progression)

Dear Prof. Ileana M Cristea:

Your manuscript has been accepted, and I am forwarding it to the ASM Journals Department for publication. For your reference, ASM Journals' address is given below. Before it can be scheduled for publication, your manuscript will be checked by the mSystems production staff to make sure that all elements meet the technical requirements for publication. They will contact you if anything needs to be revised before copyediting and production can begin. Otherwise, you will be notified when your proofs are ready to be viewed.

If you would like to submit a potential Featured Image, please email a file and a short legend to msystems@asmusa.org. Please note that we can only consider images that (i) the authors created or own and (ii) have not been previously published. By submitting, you agree that the image can be used under the same terms as the published article. File requirements: square dimensions (4" x 4"), 300 dpi resolution, RGB colorspace, TIF file format.

We recognize that the video files can become quite large, and so to avoid quality loss ASM suggests sending the video file via <https://www.wetransfer.com/>. When you have a final version of the video and the still ready to share, please send it to mSystems staff at msystems@asmusa.org.

Sincerely,

Joshua Elias
Editor, mSystems

Journals Department
E-mail: mSystems@asmusa.org